# ELODI: Ensemble Logit Difference Inhibition for Positive-Congruent Training

## Abstract

Negative flips are errors introduced in a classification system when a legacy model is updated. Existing methods to reduce the negative flip rate (NFR) either do so at the expense of overall accuracy by forcing a new model to imitate the old models, or use ensembles, which multiply inference cost prohibitively. We analyze the role of ensembles in reducing NFR and observe that they remove negative flips that are typically not close to the decision boundary, but often exhibit large deviations in the distance among their logits. Based on the observation, we present a method, called Ensemble Logit Difference Inhibition (ELODI), to train a classification system that achieves paragon performance in both error rate and NFR, at the inference cost of a single model. The method distills a homogeneous ensemble to a single student model which is used to update the classification system. ELODI also introduces a generalized distillation objective, Logit Difference Inhibition (LDI), which penalizes changes in the logits between the reference ensemble and the student single model. On multiple image classification benchmarks, model updates with ELODI demonstrate superior accuracy retention and NFR reduction.

## 1 Introduction

The rapid development of visual recognition in recent years has led to the need for frequently updating existing models in production-scale systems. However, when replacing a legacy classification model, one has to weigh the benefit of decreased error rate against the risk of introducing new errors that may disrupt post-processing pipelines (Yan et al., 2021) or cause friction with human users (Bansal et al., 2019). Positive-Congruent Training (PC-Training) refers to any training procedure that minimizes the *negative flip rate* (NFR) along with the error rate (ER).

Negative flips are instances that are misclassified by the new model, but correctly classified by the old one. They are manifest in both visual and natural language tasks (Yan et al., 2021; Xie et al., 2021). They typically include *not only* samples close to the decision boundary, but also high-confidence mistakes that lead to *perceived "regression"* in performance compared to the old model. They are present even in identical architectures trained from different initial conditions, or with different data augmentations, or using different sampling of mini-batches. Yan et al. (2021) have shown that in state-of-the-art image classification models, where a 1% improvement is considered significant, NFR can be in the order of 4∼5% even across models that have identical ER. These intriguing properties motivate us to investigate causes of negative flips and mechanism of reducing negative flips to establish a model update method that achieves the cross-model compatibility, thus lower NFR, and lower error rate, for better PC-training.

**Two questions.** A naive approach to cross-model compatibility is to bias one model to mimic the other, as done in model distillation (Hinton et al., 2015). In this case, however, compatibility comes at the expense of accuracy (Yan et al., 2021; Bansal et al., 2019). On the other hand, averaging a number of models in a deep ensemble (Lakshminarayanan et al., 2017) can reduce NFR without negative accuracy impact (Yan et al., 2021), even if it does not explicitly optimize NFR nor its surrogates. The role of ensembles in improving accuracy is widely known, but our first question arises: *what is the role of ensembles in reducing NFR?*

Even though using deep ensembles achieves state-of-the-art performance in terms of reducing NFR (Yan et al., 2021), it is not viable in real applications at scale since it multiplies the cost of

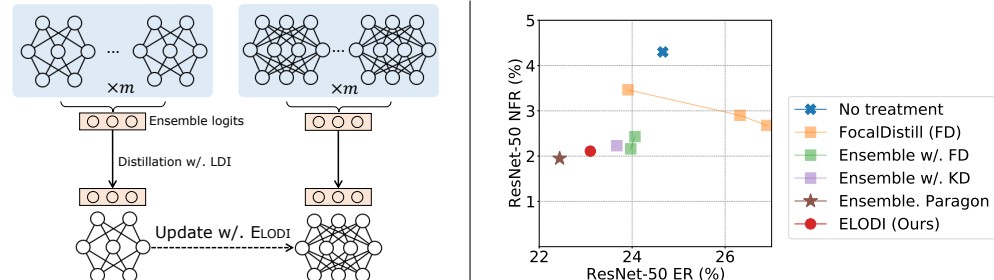

Figure 1: **Left:** In ELODI, one model is trained using the Logit Difference Inhibition (LDI) loss *w.r.t* an ensemble of $m$ models with its same architecture. The result is a single model which achieves a significantly reduced negative flip rate (NFR) with the other. **Right:** Scatter plot of a ResNet-50's ER *vs* its NFR *w.r.t* a ResNet-18. The more left and lower, the better. ELODI improves both ER and NFR than baseline methods. Particularly, ELODI is close to the ensemble paragon, without the prohibitive computation cost of ensembles.

inference by an integer factor. Therefore, a second key question arises: *Is it possible to achieve the PC-Training performance of ensembles at the inference cost of a single model?*

**Key ideas.** To address the *first* key question above, we analyze the pattern of negative flip reduction in deep ensembles. We observe that deep ensembles reduce NFR by remedying potential flip samples that have relatively large variation in the logits space of different single models. When a deep ensemble is composed of member models with the same architecture but trained with independent initialization on the same dataset, which we denote as *homogeneous ensembles*, this behavior can be theoretically predicted and empirically validated.

To address the *second* key question, we propose to train a single model by penalizing the difference of sample logits from the mean of a deep homogeneous ensemble and to use this single model to perform a model update. As illustrated in Figure 1*(Left)*, we independently train replicas of a single model with different random seeds to form the deep ensemble. We introduce a generalized distillation objective, Logit Difference Inhibition (LDI), which *only* penalizes significant changes in the logits between the reference ensemble and the student single model, to realize the ensemble to single model distillation. The result is what we call *Ensemble Logit Difference Inhibition* (**ELODI**).

**Contributions.** ELODI improves the state of the art in reducing perceived regression in model updates in three ways: (1) Generality, by not targeting distillation to a specific legacy model, yet reducing NFR; (2) Absence of collateral damage, by retaining the accuracy of a new model, or even improving it, while ensuring reduction of NFR; (3) Efficiency, as ELODI does not require evaluating ensembles of models at inference time.[1] These improvements are made possible by two main contributions: (1) an analysis on deep ensembles which sheds light on their role in reducing NFR and the direction to obtain ~~the~~ their performance for PC-training with single models; (2) ELODI, that integrates the NFR reduction of deep ensembles and running cost of single models by first training deep networks using the LDI loss with respect to an ensemble and then deploying the resulting single model at inference time. This results in a significant reduction of NFR (29% relative reduction on ImageNet for ResNet-18 → ResNet-50) over previous methods. As a side benefit, ELODI increases top-1 accuracy in several cases, and is comparable in others.

## 2 RELATED WORK

**Cross-model compatibility** is becoming increasingly important as real world systems incorporate trained components that, if replaced, can wreak havoc with post-processing pipelines. Toneva et al. (2019) empirically study prediction flip on training samples between epochs, termed "forgetting events", while Yan et al. (2021) address perceived regression using held-out sets between different models. Both are particular instance of cross-model compatibility (Shen et al., 2020; Bansal et al., 2019; Srivastava et al., 2020). Focal Distillation (Yan et al., 2021) minimizes the distance between the old and new predictions, with increased weights on samples correctly classified by the old model. Träuble et al. (2021) use a probabilistic approach to determine whether the prediction should update when a new model comes. While it improves *cumulative* NFR, it requires multiple models to be available at inference, which is prohibitive in practice.

**Ensemble learning** methods (Breiman, 1996; Freund & Schapire, 1997; Breiman, 2001) are widely adopted in machine learning. The understanding for these methods is sometimes explained as en-

---

[1]Note that ELODI is able to deal with existing models trained without treatment, as shown in Appendix B.2.

larging the margins (Bartlett et al., 1998). Recently, the "multi-view" hypothesis (Allen-Zhu & Li, 2020) suggests that each independent model in an ensemble of deep networks learns a subset of feature views and memorizes data not separable using this subset. In practice, one can always boost the performance of a classifier by averaging multiple models that are trained separately under a certain level of variation in training including model type, training data, initialization, etc. In this paper, we take a different aspect for ensembling to reduce NFR, not to improve accuracy. We apply ensemble as a teacher's model to guide the student model in reducing negative flips during model updates. In particular, we present an alternative explanation from the perspective of representations' dispersion in the logit space. ELODI can be thought of as variance reduction regularization in a Bayesian NN ensemble, which is replaced by its mean at inference time. The literature on variance reduction is too vast to survey here, but relevant references include (Hoeting et al., 1999; Fragoso et al., 2018).

**Knowledge distillation** (KD) (Hinton et al., 2015) was proposed to transfer "dark" knowledge from a larger "teacher" network to a smaller "student" by minimizing the distance between the distribution of predictions. In self distillation (Zhang et al., 2019), teacher and student are the same. Focal Distillation (Yan et al., 2021) is a special case of KD with a sample-specific filtering function, developed for model updates where the legacy "teacher" model is actually weaker than the student (new) model, as in *Reversed KD* (Yuan et al., 2020), where it is used as regularization. Ensemble distillation uses multiple teachers to improve accuracy in vision and other applications (Reich et al., 2020; Fukuda et al., 2017; Asif et al., 2020; Malinin et al., 2020; Lin et al., 2020). Our method is related to ensemble distillation while having two distinctive differences: (1) Our method uses a different term for the loss to achieve reduction of NFR; (2) our method use a *homogeneous ensemble* whose members have the same architecture and are trained on the same dataset with different initialization seeds, unlike the traditional case that uses diverse models in the ensemble (Kuncheva & Whitaker, 2003), which we call a *heterogeneous ensemble*.

## 3 REPRESENTATION LANDSCAPE OF ENSEMBLE-BASED PC-TRAINING

To answer the first key questions in Section 1, we explore (1) how negative flips occur and (2) why ensembles yield fewer negative flips. To do so, we analyze the so-called *logit space*, where the representations computed by a deep network before the softmax operation live. The reason we analyze logits rather than feature or softmax probabilities is as follows. Compared to the feature space, the logits of an arbitrary sample produced by different models, trained on the same dataset, live in the same vector space which is defined by the label set of the training samples. Compared to the space of post-softmax output, the logit distribution is easier to analyze because the softmax operation will skew the distribution. From a practical perspective, averaging in the logit space for ensembles is also common in recent works (Gontijo-Lopes et al., 2022; Wortsman et al., 2022).

### 3.1 NEGATIVE FLIPS AND LOGIT DISPLACEMENT MAGNITUDE

Given an input image $x$ and a learned model $\phi$, let $\phi(x) \in \mathbb{R}^C$ denote its output logit vector before softmax, where $C$ is the number of classes. The logit vector varies across models due to architecture, initialization, optimization method, and training dataset to name a few. For any model pair $\phi$ and $\psi$, we define the *logit displacement* to be the difference between two output logits, *i.e* $\phi(x) - \psi(x)$. Once its norm reaches a threshold and changes the order of the top predictions, a flip occurs.

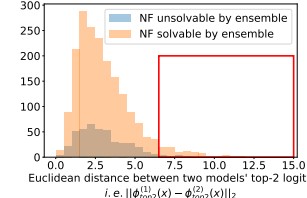

Figure 2: Negative flips with large cross-model logit displacement can be significantly reduced by ensembles. Specifically, 88 out of 97 (90.7%) negative flipped samples are reverted in the red box.

Our observation, as shown in Figure 2, is that negative flips with large cross-model logit displacement can be significantly reduced by homogeneous ensembles. In particular, we examine the Euclidean distance of a negative flipped sample's Top-2 highest output logits between two models. Replacing the single model's logits with an ensemble's, many of these negative flips can be reverted especially those with high logit distance (red box). Below we show that homogeneous ensembles indeed reduce negative flips through reducing the magnitude of logit displacement.

Let $\mathbb{M}_1 = \{\cdots, \phi^{(i)}, \cdots\}$ and $\mathbb{M}_2 = \{\cdots, \psi^{(j)}, \cdots\}$ denote the set of models of two ensembles. For simplicity, we consider $\|\mathbb{M}_1\| = \|\mathbb{M}_2\| = m$, where $\|\cdot\|$ is the cardinality of each

set. Each $\phi^{(i)}$ or $\psi^{(i)}$ has the same model architecture, respectively, and are trained on the same dataset, despite being independently initialized so that they are homogenous ensembles. Given an input image $x$, $\{\phi^{(1)}(x), \cdots, \phi^{(m)}(x)\}$ can be considered as $m$ *i.i.d* random variables drawn from a distribution approximated to second-order by an expectation $\boldsymbol{\mu}_1$ and a co-variance matrix $\boldsymbol{\Sigma}_1$, *i.e* $\phi(x) \sim \mathcal{D}(\boldsymbol{\mu}_1, \boldsymbol{\Sigma}_1)$. Likewise $\psi(x) \sim \mathcal{D}(\boldsymbol{\mu}_2, \boldsymbol{\Sigma}_2)$. The ensembles' logit vectors are computed as $\phi^{(\mathrm{ens})}(x)$ and $\psi^{(\mathrm{ens})}(x)$ by averaging the individual models' logits. The multi-dimensional central limit theorem (CLT) (Rvačeva, 1962; Ferguson, 2017) states that this average converges in distribution to a multivariate normal distribution with the increase of $m$, *i.e*

$$\phi^{(\mathrm{ens})}(x) = \frac{1}{m}\sum_{i=1}^{m}\phi^{(i)}(x) \overset{D}{\sim} \mathcal{N}(\boldsymbol{\mu}, \frac{1}{m}\boldsymbol{\Sigma}). \tag{1}$$

Therefore, the logit displacement between the two ensembles converges in distribution to another multivariate normal distribution, *i.e*

$$\phi^{(\mathrm{ens})}(x) - \psi^{(\mathrm{ens})}(x) = \frac{1}{m}\sum_{i\in\mathbb{M}_1}\phi^{(i)}(x) - \frac{1}{m}\sum_{j\in\mathbb{M}_2}\psi^{(j)}(x) \overset{D}{\sim} \mathcal{N}\left(\boldsymbol{\mu}_1 - \boldsymbol{\mu}_2, \frac{\boldsymbol{\Sigma}_1 + \boldsymbol{\Sigma}_2}{m}\right). \tag{2}$$

The norm of logit displacement will follow a generalized $\chi^2$ distribution (Mathai & Provost, 1992; Das & Geisler, 2021).

As a special case, if $\mathbb{M}_1$ and $\mathbb{M}_2$ have the same model architecture, then we have a normal distribution with *zero* mean and co-variance inversely scaled by the ensemble size:

$$\phi^{(\mathrm{ens})}(x) - \psi^{(\mathrm{ens})}(x) \overset{D}{\sim} \mathcal{N}\left(\mathbf{0}, \frac{\boldsymbol{\Sigma}_1 + \boldsymbol{\Sigma}_2}{m}\right). \tag{3}$$

We connect the analysis to the observations shown in Figure 4: (1) Equation 3 implies that when ensembles become larger, the expectation of logit difference is zero and the covariance keeps decreasing, resulting in consistently decreasing NFR. This is consistent with the observation in (Yan et al., 2021), which is redrawn in the red curve, that two very large ensembles with the same architecture can have almost no flips. (2) In the case of two homogeneous ensembles with different architectures, $\boldsymbol{\mu}_1$ and $\boldsymbol{\mu}_2$ could have a non-zero difference which results in NFR not converging to zero. But the decrease of covariance part still contributes to consistent non-trivial reduction in NFR. These explain the observation in the orange curve that NFR stagnates at a smaller non-zero value. Additionally, although we cannot conduct similar analysis for a heterogeneous ensemble, we can empirically verify that the NFR between two heterogeneous ensembles has significantly higher variance, as shown in the purple curve, which suggests they may be inferior for NFR reduction.

## 3.2 VALIDATING THE HYPOTHESIS OF REDUCING LOGIT DISPLACEMENT MAGNITUDE

We validate our hypothesis on the representation landscape through large-scale experiments. Specifically, we train 256 ResNet-18 models on ImageNet with different seeds and split them into two halves. For an arbitrary image, we randomly draw $m$ models without replacement in each half and compute the averaged logits of this drawn ensemble. We repeat the process and present in Figure 3a the histogram of the logit displacement's $\ell_2$ norm between two random ensembles. Note that Seguin et al. (2021) argue that the logit distribution is highly affected by the number of training epochs, therefore we follow the standard training recipe in Appendix A for its assumption to hold.

We examine our hypothesis in Section 3.1 by comparing the histogram with the probability mass function (PMF) of logit difference norm. We start by using all available single models to estimate mean and co-variance $(\boldsymbol{\mu}, \boldsymbol{\Sigma})$ for the logit vectors' distribution $\phi$ and examine whether the norm of logit displacement will follow a generalized (central) $\chi^2$ distribution. Since the probability density function (PDF) of a generalized $\chi^2$ variable does not have a simple closed-form expression, we estimate it by Kernel Density Estimation (KDE) method (Parzen, 1962). From Figure 3a, we see that the simulated PMFs in solid lines fit the histogram of single models well, implying that logits of these models could indeed follow a normal distribution. We conduct the same experiments above on more images in Appendix E and the conclusion holds well, suggesting this property is not incidental. If we move to the ensemble case, the logit displacement follows another normal distribution with scaled co-variance matrix, which leads to another generalized $\chi^2$ distribution with shifted mean and scaled covariance. Its estimated PMF, shown in dashed lines in Figure 3a, is consistent with the corresponding histogram.

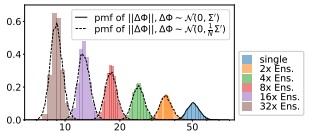 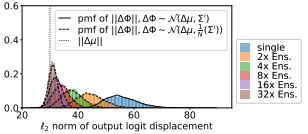 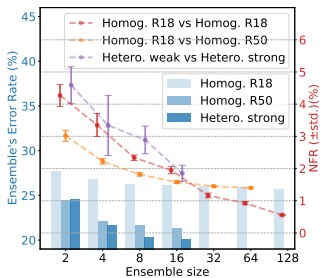

(a) Homog. R-18 Ensemble *vs* Homog. R-18 Ensemble. $\Delta\boldsymbol{\mu} = \mathbf{0}, \boldsymbol{\Sigma}' = 2\boldsymbol{\Sigma}_{1,2}$.

(b) Homog. R-18 Ensemble *vs* Homog. R-50 Ensemble. $\Delta\boldsymbol{\mu} \neq \mathbf{0}, \boldsymbol{\Sigma}' = \boldsymbol{\Sigma}_1 + \boldsymbol{\Sigma}_2$.

Figure 3: $\ell_2$ **norm histogram of logit displacement between two random ensembles.** The bin size is 0.5. We also plot the simulated PMFs: solid lines for $\ell_2$ norm of a simulated normal distribution $\mathcal{N}\left(\Delta\boldsymbol{\mu}, (\boldsymbol{\Sigma}_1 + \boldsymbol{\Sigma}_2)\right)$ whose parameters are estimated from all available single models; dashed lines for those of extrapolated distribution $\mathcal{N}\left(\Delta\boldsymbol{\mu}, \frac{1}{m}\boldsymbol{\Sigma}'\right)$. Consistency between the histograms and PMFs supports our hypotheses in Section 3.1.

Figure 4: **ER/NFR *w.r.t* ensemble size**. The behaviors of model updates under different ensemble settings can be explained by our hypothesis. Please refer to the last paragraph in Section 3.1.

---

**Algorithm 1:** Sequential model update with ELODI

**Data:** Dataset $\mathbb{X}$; number of versions to update $T$
**Result:** A sequence of models $\hat{\boldsymbol{\phi}}^{(\text{Ver-0})}, \cdots, \hat{\boldsymbol{\phi}}^{(\text{Ver-}T)}$
$t \leftarrow 0$;                                    `/* initial version */`
**repeat**
     **for** $i = 1 \cdots m$ **do**
         $\boldsymbol{\phi}^{(i,\text{Ver-}t)} \leftarrow \arg\min \sum_{x \in \mathbb{X}} \mathcal{L}_{\text{CE}}\left(\boldsymbol{\phi}^{(\text{Ver-}t)}\right)$;         `/* Train m models */`
     **end**
     **for** $x \in \mathbb{X}$ **do**
         $\boldsymbol{\phi}^{(\text{ens,Ver-}t)}(x) \leftarrow \frac{1}{m}\sum_{i \in \{1,\cdots,m\}} \boldsymbol{\phi}^{(i,\text{Ver-}t)}(x)$;    `/* Build old ensemble */`
     **end**
     $\hat{\boldsymbol{\phi}}^{(\text{Ver-}t)} \leftarrow \arg\min_{\boldsymbol{\phi}} \sum_{x \in \mathbb{X}} \mathcal{L}_{\text{distill}}\left(\boldsymbol{\phi}(x), \boldsymbol{\phi}^{(\text{ens,Ver-}t)}(x)\right)$;     `/* Distill into a`
     `single model and publish */`
**until** $t \geq T$;

---

The sample-specific logit displacement norm between two homogeneous ensembles with different architectures can be analyzed likewise. Considering $\boldsymbol{\phi}(x) \sim \mathcal{N}(\boldsymbol{\mu}_1, \boldsymbol{\Sigma}_1)$ for ResNet-18 and $\boldsymbol{\psi}(x) \sim \mathcal{N}(\boldsymbol{\mu}_2, \boldsymbol{\Sigma}_2)$ for ResNet-50, the norm of logit displacement should follow a generalized *non-central* $\chi^2$ distribution. From Figure 3b we can see that the estimated distribution of the logit displacement norm fits the empirical distribution well. It still condenses as the ensemble get larger but much slower than Figure 3a. Also the mean converges to non-zero (the dotted vertical line in Figure 3b).

## 4 METHOD: ENSEMBLE LOGIT DIFFERENCE INHIBITION (ELODI)

The above analysis suggests the effectiveness of large homogeneous ensembles in reducing NFR, but an ensemble is less practical compared with a single model due to its multiplied inference cost to run every member model on a new input. In this work, we propose to re-purpose the knowledge distillation technique (Hinton et al., 2015), which was previously used for improving model accuracy, for the task of transferring NFR reduction capability from ensembles to single models.

### 4.1 UPDATES WITH ENSEMBLE-DISTILLED MODELS

Given an ensemble $\boldsymbol{\phi}^{(\text{ens})}$ composed of a set of models $\mathbb{M} = \{\boldsymbol{\phi}^{(i)}\}_{i=1}^{m}$, we learn a *single* model $\hat{\boldsymbol{\phi}}$ such that for each sample $x \in \mathbb{X}$, the random variable of sample logits can have ensemble-like reduced variance. We then use single models learned in this way in every model update of a system.

As illustrated in Figure 1*(Left)*, when an old model $\hat{\boldsymbol{\phi}}^{(\text{old})}$ needs to be updated to a new model, we first train a homogeneous ensemble on the same dataset and having the same architecture as the desired new model. Next we distill this reference ensemble to the actual new model $\hat{\boldsymbol{\phi}}^{(\text{new})}$. If the distillation process is able to convey the property of reducing NFR to the learned single model, updating from $\hat{\boldsymbol{\phi}}^{(\text{old})}$, which was preferably produced in the same manner, to $\hat{\boldsymbol{\phi}}^{(\text{new})}$ would result in significantly reduced NFR than that of a model pair without this treatment.

We empirically verify this model update method, called ELODI, in Section 5.1 and find it effective in reducing NFR while retaining the accuracy gain introduced by the new architecture used in the new model. It avoids the prohibitive inference cost of deep ensembles due to only using $\hat{\phi}^{(\text{new})}$ in the model update, with reference ensemble discarded after the distillation. Another benefit of ELODI is that new model does not need to target any specific existing model in an update, enabling a chain of models to yield relatively low NFR between any pair of them. This is rather helpful when multiple updates are consecutively executed, which is common for a long-running machine learning system. The overall pipeline of a sequential model update via ELODI is summarized in Algorithm 1.

## 4.2 LOSS CHOICES OF ELODI

Generally we obtain the model to be deployed using the distillation technique (Hinton et al., 2015),

$$\hat{\phi} = \arg\min_{\phi} \sum_{x \in \mathbb{X}} \mathcal{L}_{\text{distill}} \left( \phi(x), \phi^{(\text{ens})}(x) \right). \tag{4}$$

Various types of distilling function have been proposed to improve single-model accuracy: (1) The vanilla KD loss (Hinton et al., 2015) minimizes the KL-divergence between two models' output logits; (2) FitNet (Romero et al., 2015) and AttentionTransfer (Zagoruyko & Komodakis, 2017) mimics the intermediate hidden layer's activation or attention; (3) FSP (Yim et al., 2017) mimics the cross-layer Gram matrices. Our goal is to minimize the norm of covariance matrix in terms of sample logits between the learned model and the ensemble,

$$\hat{\phi} = \arg\min_{\phi} \sum_{x \in \mathbb{X}} \|\text{Var}(\phi(x)) - \text{Var}(\phi^{(\text{ens})}(x))\|. \tag{5}$$

However, computing either term in Equation 5 is hard: (1) $\text{Var}(\phi(x))$ is ill-posed while training a single model; (2) So is $\text{Var}(\phi^{(\text{ens})}(x))$ because the matrix dimension $C$, *i.e* the number of classes, is usually by several orders of magnitude larger than the ensemble size $m$ (a typical value in the experiments is $C = 1,000$ and $m = 8$). As an alternative, we propose a new distillation loss named logit difference inhibition (LDI) that focuses on the first-order statistics but achieves a similar goal of variance reduction. It can be viewed as a generalized version of KD loss but is more focused on penalizing large logit displacement towards the ensemble rather than exact logit matching.

Let $\phi_k^{(\cdot)}(x)$ denote the $k$-th element of the logit vector. Motivated by the analysis in Section 3.1, instead of exact match for boosting accuracy, we only *inhibit* significant difference of logit vectors' elements between the distilled model and the ensemble as follows:

$$\mathcal{L}_{\text{LDI}}(x) = \sum_{k \in \mathbb{K}} \max \left( 0, \|\phi_k(x) - \phi_k^{(\text{ens})}(x)\| - \xi \right)^p, \tag{6}$$

$\xi$ is a non-negative truncating threshold such that difference below $\xi$ is truncated. $\xi$ is set to be 0.1. $\mathbb{K}$ is a data-dependent subset of classes where the loss will be applied. $\mathbb{K}$ can include either all classes ($\{1, \cdots, C\}$) or the classes with the top-$K$ highest logit elements. This top-$K$ design choice is motivated by our observation that flips are likely to happen in classes with high logit displacements. As shown in experiments, the Top-$K$ variant leads to no loss in NFR reduction compared to the full form. It could instead help in reducing computation cost when number of classes are extremely large (An et al., 2020). $p$ is 2 in our experiments. When $\xi = 0$ and $\mathbb{K} = \{1, \cdots, C\}$, LDI degrades to the logit matching loss (Hinton et al., 2015).

**The overall objective** of the distillation in ELODI is a weighted sum of standard Cross Entropy and the LDI loss, *i.e* $\mathcal{L} = (1 - \alpha)\mathcal{L}_{\text{CE}} + \alpha\mathcal{L}_{\text{LDI}}$, where the weight $\alpha$ is set such that the magnitude of $\mathcal{L}_{\text{CE}}$ and $\mathcal{L}_{\text{LDI}}$ is similar.

## 5 EXPERIMENTS

We validate the proposed approaches on two standard image classification datasets: ImageNet (Deng et al., 2009) and iNaturalist (Van Horn et al., 2018). For a new model in an update experiment, we measure top-1 error rate (**ER**) and its negative flip rate to the old model (**NFR**) (Yan et al., 2021), computed as $\text{NFR} = \frac{1}{N} \sum_{i=1}^{N} \mathbb{I}(\hat{y}_i^{(\text{new})} \neq \ell_i, \hat{y}_i^{(\text{old})} = \ell_i)$, where $\mathbb{I}(\cdot)$ is the indicator function, $\ell_i$ is the label, and $\hat{y}_i^{(\text{new})}$ ($\hat{y}_i^{(\text{old})}$) is the new (old) model's prediction. The training recipe is described in Appendix A. Unless otherwise specified, ELODI experiments are done with ensemble size $m = 8$.

Table 1: **Comparing ELODI with other PC-Training Methods.** $^*$ in ensemble paragon means that the number is from a collection of models. The middle rows are targeted model update baselines, where the old ResNet-18 is referenced when training the new ResNet-50. The bottom rows are ELODI and its Top-$K$ variants.

| Method | ER$_\downarrow$(%) | | NFR$_\downarrow$(%) |
|---|---|---|---|
| | ResNet-18 | ResNet-50 | |
| No treatment (single) | 30.24 | 24.66 | 4.30 |
| Ensemble Paragon (8×) | 26.34* | 22.44* | 1.95 |
| (TARGETED MODEL UPDATE) | | | |
| BCT (Shen et al., 2020) | 30.24 | 25.00 | 4.34 |
| KD (Hinton et al., 2015) | — " — | 28.38 | 3.20 |
| FD-KL (Yan et al., 2021) | — " — | 26.32 | 2.90 |
| BU-CR (Träuble et al., 2021) | — " — | 26.51 | 4.56 |
| ELODI | 31.34 | 23.15 | 2.18 |
| Top-$K$ ELODI ($K = 10$) | 30.95 | 23.10 | 2.11 |

Table 2: **ELODI in data-growth settings on ImageNet.** ◖(○) means that model is trained on half (full) data.

| Method | Increasing #classes | | | | Increasing #samples/class | | |
|---|---|---|---|---|---|---|---|
| | Error Rate$_\downarrow$(%) | | NFR$_\downarrow$ | | Error Rate$_\downarrow$(%) | | NFR$_\downarrow$ |
| | R18◖ | R50○ | | | R18◖ | R50○ | |
| No treatment | 22.02 | 24.66 | 14.07 | | 34.26 | 24.66 | 3.52 |
| FD (Yan et al., 2021) | — " — | 39.96 | 5.45 | | — " — | 33.06 | 2.65 |
| Ensemble Paragon (8×) | 18.70* | 22.44* | 4.12 | | 29.16* | 22.44* | 2.11 |
| ELODI | 21.80 | 23.15 | 4.19 | | 34.08 | 23.15 | 2.25 |

## 5.1 MAIN RESULTS OF ELODI

**Comparison with other PC-Training Methods.** We first compare ELODI with other PC-Training methods under the setting of updating ResNet-18 to ResNet-50. "No treatment" means that both ResNet-18 and -50 are trained with standard cross entropy loss. We can see that ELODI outperforms previous methods such as Backward-Compatible Training (BCT) (Shen et al., 2020), Knowledge Distillation (KD) (Hinton et al., 2015), Focal Distillation (FD-KD/LM) (Yan et al., 2021) and Bayesian Update (BU) (Träuble et al., 2021)[2]. Note that these baselines are all *targeted model update*, where the old ResNet-18 is used as target when training ResNet-50 while ELODI does not target at any legacy model. Second, we can apply the LDI loss in the targeted model update setting and it achieves slightly worse NFR compared to FD while attaining ResNet-50's accuracy significantly better. See more discussion in Section 5.3. Third, we study a few variants of ELODI. Using the Top-$K$ highest-logit class subset in ELODI with $K \in \{2, 5, 10\}$ does not deteriorate the performance ($\pm 0.1\%$). From an ER-NFR scatter plot in Figure 1*(Right)*, ELODI achieves a similar level of ER-NFR results as the ensemble paragon (Yan et al., 2021) at the inference cost of a single model.

## 5.2 MORE UPDATE SETTINGS OF ELODI

**Data-growth settings.** Model updates may also come with the growth of training data, *e.g* (1) increasing number of classes, and (2) increasing number of per-class samples. We follow the same data/class split in (Yan et al., 2021), which uses 50% classes/samples for old model and full data for new. From Table 2, we find that conclusions from the full-data setting also hold for these settings.

**Fine-tuning on other datasets.** We validate the effectiveness of ELODI when transferring to iNaturalist (Van Horn et al., 2018) following the protocol in (Yan et al., 2021). Results of both full-data and data-growth setting are summarized in Table 3, where ELODI consistently outperforms FD.

**ELODI on a chain of model updates.** To illustrate Algorithm 1, we show the *transitivity* of NFR reduction induced by ELODI in chain updates of three models, *i.e* ResNet-18→ResNet-50→ResNet-101. As shown in Table 6, with ELODI, NFRs between the three models reduce to $2.04\% \sim 2.25\%$ from $3.92\% \sim 4.41\%$ (a relative reduction of $44.1\% \sim 52.3\%$). Note this is achieved without

---

[2]In our re-implementation, we use CostRatio(CR)-based Bayesian update rule to get the target and train the new model online instead of offline model update in the original paper.

Table 3: **ELODI on iNaturalist.** ELODI is effective when fine-tuning on iNaturalist under multiple settings.

| Method | Increasing #classes | | | Increasing #samples/class | | | Full data | | |
|---|---|---|---|---|---|---|---|---|---|
| | ER$_\downarrow$(%) | | NFR$_\downarrow$ | ER$_\downarrow$(%) | | NFR$_\downarrow$ | ER$_\downarrow$(%) | | NFR$_\downarrow$ |
| | R18$^\bullet$ | R50$^\circ$ | | R18$^\bullet$ | R50$^\circ$ | | R18$^\bullet$ | R50$^\circ$ | |
| No treatment | 47.58 | 35.95 | 5.38 | 78.88 | 35.95 | 3.82 | 40.69 | 35.95 | 4.76 |
| FD (Yan et al., 2021) | — " — | 45.87 | 3.44 | — " — | 66.91 | 2.00 | — " — | 40.03 | 3.95 |
| Ensemble Paragon (8×) | 36.03* | 29.47* | 1.68 | 47.04* | 29.47* | 1.23 | 36.25* | 29.47* | 2.10 |
| ELODI | 43.56 | 34.29 | 1.91 | 52.96 | 34.29 | 1.47 | 40.37 | 34.29 | 2.46 |

Table 4: **ELODI with different guiding ensembles.** We consider ResNet-18 →ResNet-50 via ELODI with an 8×-model ensemble. **All-diff-weak (-strong)** ensemble is composed of 8 different weak (strong) models with Top-1 Acc. $\approx 69\%$ (75%) on ImageNet. The model list can be found in Appendix C. **Mixed-weak (-strong)** ensemble is a mixture of 4× ResNet-18 and 4× VGG-13 (4× ResNet-50 and 4× DenseNet-121).

| Old Reference | New Reference | Error Rate$_\downarrow$ (%) | | NFR$_\downarrow$ (%) |
|---|---|---|---|---|
| | | R18$^\diamond$ | R50$^\diamond$ | R18$^\diamond$→R50$^\diamond$ |
| N/A | N/A | 30.24 | 24.66 | 4.30 |
| All-diff-weak | All-diff-weak | 32.38 | 26.11 | 1.94 |
| Mixed-weak | Mixed-weak | 32.75 | 26.88 | 1.99 |
| R-18 (×8) | R-18 (×8) | 31.32 | 26.82 | 2.06 |
| All-diff-weak | All-diff-strong | 32.73 | 23.14 | 2.25 |
| All-diff-weak (calibrated) | All-diff-strong (calibrated) | 33.38 | 23.33 | 2.35 |
| Mixed-weak | Mixed-strong | 32.75 | 23.68 | 2.24 |
| R-18 (×8) | R-50 (×8) | 31.32 | 23.15 | 2.18 |

Table 5: **ELODI with different architectures on ImageNet**. ELODI effectively reduces NFR on a wide range of architectures. $^\dagger$ is obtained by our reproduction with different augmentation and training schedule. Note that all new models' NFR is measured *w.r.t* ResNet-18 listed in the leftmost column.

| | ER$_\downarrow$ (%) R-18 (old) | ER$_\downarrow$ (%) R-101 | NFR$_\downarrow$ (%) →R-101 | ER$_\downarrow$ (%) D-161 | NFR$_\downarrow$ (%) →D-161 | ER$_\downarrow$ (%) SwinT | NFR$_\downarrow$ (%) →SwinT |
|---|---|---|---|---|---|---|---|
| None (single) | 30.24 | 24.66 | 3.64 | 21.82 | 3.73 | 20.40$^\dagger$ | 3.77 |
| Ensemble (8×) | 26.34* | 20.05* | 1.72 | 18.90* | 2.06 | 18.37* | 2.60 |
| ELODI | 31.34 | 21.09 | 2.19 | 21.74 | 2.57 | 19.84 | 2.95 |

crafting the complex reference schemes which are necessary for baselines since they require old models' references. The details of other baseline methods are discussed in Appendix B.1.

**Updates to dissimilar architectures.** In Table 5, we study if ELODI is applicable to updates across dissimilar architectures (ResNet-18→DenseNet-161/Tiny-Swin) besides similar ones (ResNet-101). We see that ELODI effectively reduces NFR in all cases, with retained or sometimes decreased ER.

## 5.3 DESIGN CHOICES OF ELODI

**Homogeneous *vs* heterogeneous ensembles**. In both Section 3.1 and 4.1, we use homogeneous ensemble. However, in ensemble learning, members with strong diversity such as model architectures are usually favored for better generalization. In Table 4, we observe that using homogeneous ensemble for guidance achieves comparable or slightly better results in both NFR and ER than heterogeneous ("all-different") ensembles. We also additionally calibrate "all-different" ensembles (Guo et al., 2017) but observe no gain. This suggests that strong diversity in a guiding ensemble may not lead to lower NFR. Also, ELODI with homogeneous ensembles is easier to implement and extend.

**Change of architecture for the guiding ensemble**. In Table 4, we find that training a new model guided by an ensemble with the old model's architecture has to trade ER for reduction of NFR, which is not desired. This corroborates with the hypothesis in Section 3.1 that models with different architectures has different representation landscape and thus it is better to use ensemble with the same architecture for guiding ELODI. When a system has gone through multiple updates, always guiding ELODI with the new model's architecture also provides a clear guideline for practice.

Table 6: **ELODI achieves lower Pairwise NFR on a chain of models.** $\mathcal{M}_1 \rightarrow \mathcal{M}_2$ means that we measure $\mathcal{M}_2$'s NFR *w.r.t* $\mathcal{M}_1$. $\mathcal{M}^\diamond$ means that $\mathcal{M}$ is trained from ELODI.

| Method | Pairwise NFR |
|--------|--------------|
| None | R18 $\xrightarrow[4.28\%]{3.92\%}$ R50 $\xrightarrow{4.41\%}$ R101 |
| ELODI | R18$^\diamond$ $\xrightarrow[2.04\%]{2.19\%}$ R50$^\diamond$ $\xrightarrow{2.25\%}$ R101$^\diamond$ |

Table 7: **Distilling ensembles with different loss functions**. Considering model update of ResNet-18→ResNet-50, ELODI achieves lower NFR and ER than KD/FD.

| Method | ER$_\downarrow$(%) | | NFR$_\downarrow$(%) |
|--------|---------|--------|---------|
| | R18$^\diamond$ | R50$^\diamond$ | |
| ELODI | 31.0 | 23.28 | 2.26 |
| Ensemble w/. KD$_{\tau=100}$ | 32.09 | 23.67 | 2.23 |
| Ensemble w/. FD$_{\tau=100}$ | 32.19 | 23.97 | 2.16 |
| Ensemble w/. FD$_{\tau=1}$ | 31.62 | 24.06 | 2.43 |

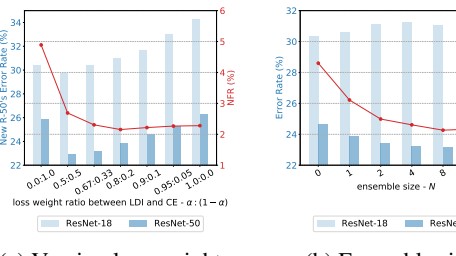

(a) Varying loss weights.  (b) Ensemble size.

Figure 5: **Ablating loss weights and ensemble sizes for ELODI on ImageNet.** ER of ResNet-18 (-50) is shown in the light (dark) blue bar plot while NFR in the red curve.

Table 8: **Comparison between offline and online distill on ImageNet.** Inferring teacher logits during training (online) achieves both lower ER and NFR compared to pre-extracting it (offline).

| Method | Error Rate$_\downarrow$(%) | | NFR$_\downarrow$(%) |
|--------|---------|--------|---------|
| | R18$^\diamond$ | R50$^\diamond$ | |
| Offline | 32.49 | 24.26 | 2.38 |
| Online | 30.97 | 23.81 | 2.15 |

**Choices of distillation functions**. We compare different choices of distillation loss functions in Table 7. We can see that using LDI loss, *i.e* ELODI, outperforms FD (Yan et al., 2021) (Ensemble w/. FD) or KD (Hinton et al., 2015) (Ensemble w/. KD) in both ER and NFR.

## 5.4 EXPLORATORY STUDIES OF PARAMETERS IN ELODI

**The effect of loss weight.** We experiment with different loss weight $\alpha$ and summarize the results in Figure 5a. $\alpha_{\text{ELODI}} = 0$ is equivalent to the no-treatment baseline. When $\alpha_{\text{LDI}}$ increases from 0.5 to 1, the distilled model's ER first decreases and then increases for both models. On the other hand, NFR consistently decreases and stays at ~2.2%. We find $\alpha_{\text{ELODI}} = 0.8$ achieves a good balance between the distilled model's ER and NFR. Therefore we use it by default for all ELODI experiments.

**The size of reference ensemble.** We study ELODI's efficiency for reducing NFR by varying the ensemble size $m$ in Figure 5b. The case of $m = 0$ is the no-treatment baseline. The case of $m = 1$ can be viewed as self distillation (Zhang et al., 2019) except that the new model's weight is re-initialized with a different random seed. NFR decreases from 4.30% to 2.15% by when the ensemble size increases from 1 to 8.

**Online *vs* offline distillation.** In ELODI, the ensemble's logits can be either inferred during training (online) or pre-extracted before training (offline). In Table 8, we find that offline distillation is less effective in reducing NFR and ER. Similar observation is also reported in (Beyer et al., 2022). Therefore we use the online approach in all experiments.

## 6 DISCUSSION

Our experiments show that ELODI performs positive congruent training by reducing negative flips with large logit displacement and reducing the variance of logits from the ensemble estimates. This behavior can be transferred to single models through the ELODI method can benefit updates with single models. As discussed in Section 3.1 and later observed in experiments, the difference in representation landscape could still lead to non-zero NFR in the updates even with ELODI, which requires future works on in-depth characterizing of representation landscape change between model architectures. Another limitation of ELODI is that the training cost is still higher than the normal training process of a classification model update, due to the additional training of the ensemble and online inference of the ensemble logits, calling for further efficiency improvement.

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

## A  Training Recipe

We follow standard training recipe of ImageNet[3]. Specifically, all classification models are trained with SGD with momentum of 0.9 and base learning rate of 0.1, which is reduced by $\frac{1}{10}$ every 30 epochs until 90 epochs. The batchsize is 256 with 8 GPUs.

For ELODI with larger models and ensemble size (*e.g* $8\times$ DenseNet-161), GPU memory becomes a bottleneck. To handle the memory issue, we use gradient checkpointing (Chen et al., 2016) and reduce batch size while linearly scaling base learning rate (Goyal et al., 2017).

## B  More Experiments on Elodi

### B.1  Elodi on a chain of model updates

As discussed in Section 4.1, ELODI does not involve the guiding ensembles at the inference stage. Only the single model trained with ELODI is deployed to replaced the old model, which is also trained with ELODI. When the number of updates increase, this naturally forms a chain of models having low NFR between them.

We illustrate this *transitivity* of NFR reduction induced by ELODI in chain updates of three models, *i.e* ResNet-18→ResNet-50→ResNet-101. As shown in Table 9a, with ELODI, the NFR between the three models tested reduced to $2.04\% \sim 2.25\%$ from $3.92\% \sim 4.41\%$ (a relative reduction of $44.1\% \sim 52.3\%$), outperforming all previous methods, including variants of FD (Yan et al., 2021) and LDI. Note this is achieved without crafting the complex reference schemes which are necessary for baselines since they require old models' references.

### B.2  Integrating Elodi with existing models

**Dealing with old models without ELODI** is necessary when updating an existing system. We consider the simple case of one old model not trained with ELODI. In this case, we augment ELODI with an additional LDI loss *w.r.t* to the old model, *i.e*

$$\mathcal{L}_{\text{total}} = \lambda \mathcal{L}_{\text{ELODI}}(\mathcal{M}_{\text{new}}^{\diamondsuit}; \mathcal{M}_{\text{new}}^{(\text{ens})}) + (1 - \lambda)\mathcal{L}_{\text{LDI}}(\mathcal{M}_{\text{new}}^{\diamondsuit}; \mathcal{M}_{\text{old}}), \tag{7}$$

where $\mathcal{M}^{\diamondsuit}$ denote the model to be guided from the ensemble.

To show some empirical results, here we consider three models, ResNet-18→ResNet-50→ResNet-101, where ResNet-18 model is trained without ELODI. Therefore both ResNet-50$^{\diamondsuit}$ and -101$^{\diamondsuit}$ will have a higher NFR compared with ResNet-18. To handle this, we introduce an additional LDI loss targeted at ResNet-18 when training ResNet-50 and (or) ResNet-101 using ELODI. The results are shown in Table 9b. We can see that ELODI w/. LDI outperforms ELODI w/o. LDI on all pairwise NFRs, indicating that augmenting ELODI with LDI loss towards the existing model is effective in dealing with this legacy case.

## C  Details of Heterogeneous Ensembles in Figure 4 and Table 4

**Model list of heterogeneous ensembles.** The **all-diff-weak** ensemble is composed of 8 different weak models with Top-1 Accuracy $= 69 \sim 70\%$ on ImageNet, including: ResNet-18 (He et al., 2016), GoogleNet (Szegedy et al., 2015), VGG-11, VGG-13, VGG-11-BN, VGG-16 (Simonyan & Zisserman, 2014; Ioffe & Szegedy, 2015), HRNet-W18 (Wang et al., 2020), DLA-34 (Yu et al., 2018). The **all-diff-strong** ensemble is composed of 8 different strong models with Top-1 Accuracy $= 75 \sim 76\%$ on ImageNet, including ResNet-50 (He et al., 2016), DenseNet-121 (Huang et al., 2017), Inception-V3 (Szegedy et al., 2016), VGG-19-BN (Simonyan & Zisserman, 2014), RegNetY (Radosavovic et al., 2020), RepVGG-A2 (Ding et al., 2021), DPN-68 (Chen et al., 2017), DLA-X-60-C (Yu et al., 2018). All of the model weights are adopted from PyTorch Image Models (TIMM) (Wightman, 2019).

---

[3]https://github.com/pytorch/examples/tree/main/imagenet

Table 9: **Pairwise NFR on multiple models.** $\mathcal{M}_1 \to \mathcal{M}_2$ means that we measure $\mathcal{M}_2$'s NFR *w.r.t* $\mathcal{M}_1$. $\mathcal{M}_1 \Rightarrow \mathcal{M}_2$ means that $\mathcal{M}_2$ is trained with $\mathcal{M}_1$ being teacher using distillation loss, *e.g* FD or LDI. $\mathcal{M}^\diamond$ means that $\mathcal{M}$ is trained from ELODI.

| Method | Pairwise NFR | | Method | Pairwise NFR | |
|---|---|---|---|---|---|
| None | R18 $\to$ R50 $\to$ R101 (3.92%, 4.28%, 4.41%) | | ELODI | R18$^\diamond$ $\to$ R50$^\diamond$ $\to$ R101$^\diamond$ (**2.19%**, **2.04%**, 2.25%) | |
| FD (chain) | R18 $\Rightarrow$ R50 $\Rightarrow$ R101 (3.46%, 2.90%, 2.13%) | | LDI (chain) | R18 $\Rightarrow$ R50 $\Rightarrow$ R101 (2.69%, 2.86%, 2.35%) | |
| FD (radial) | R18 $\Rightarrow$ R50 $\Rightarrow$ R101 (2.63%, 2.90%, 2.33%) | | LDI (radial) | R18 $\Rightarrow$ R50 $\Rightarrow$ R101 (2.57%, 2.86%, 3.09%) | |
| FD (fc) | R18 $\Rightarrow$ R50 $\Rightarrow$ R101 (2.96%, 2.90%, **1.97%**) | | LDI (fc) | R18 $\Rightarrow$ R50 $\Rightarrow$ R101 (2.68%, 2.86%, 2.96%) | |

(a) Sequential update. (1) **chain**: each model targets at its closest predecessor; (2) **radial**: each model targets at its farthest ancestor; (3) fully-connected (**fc**): each model targets at all its ancestors.

| LDI usage | Pairwise NFR |
|---|---|
| None | R-18 $\to$ R-50$^\diamond$ $\to$ R-101$^\diamond$ (2.56%, 2.98%, 2.25%) |
| Once | R-18 $\Rightarrow$ R-50$^\diamond$ $\to$ R-101$^\diamond$ (2.56%, **2.85%**, 2.14%) |
| Both | R-18 $\Rightarrow$ R-50$^\diamond$ $\Rightarrow$ R-101$^\diamond$ (**2.48%**, **2.85%**, **2.12%**) |

(b) Integrating ELODI with existing models.

**Calibration of heterogeneous ensembles.** We use temperature scaling (Guo et al., 2017) for each individual model in the ensemble set to make the confidence score match the true correctness likelihood on validation set. The temperature for scaling is optimized based on the loss on validation set with LBFGS. Since each model in the ensemble set has similar accuracy, after such calibration, their confidence scores are matched to the same level.

# D   A TWO-DIMENSIONAL EXAMPLE

To illustrate the behavior of models in logit space, we create a toy example by selecting *two* classes[4] from ImageNet (Deng et al., 2009) and training several ResNet-18 models for *binary* classification. The models differ by their initialization, determined by distinct random seeds; we then collect output logits for each test datum and model in the ensemble. In Figure 6a, we plot the two-dimensional logit vectors of multiple data points when updating from an individual model to another. We can roughly categorize the negative flipped samples, highlighted with the purple arrows, into two types: (1) those close to the decision boundary in the old model; (2) those far from the decision boundary in the old model but still flipped in the new one, due to significant displacement of the logit vector. Figure 6b shows the logit vectors of the same set of data points but in the update case of two ensemble models each having 3 members ($3\times$). Compared to Figure 6a, we can observe a clear reduction in the magnitude of displacement during the update. To validate that this observation is not incidental, we construct many cases of model updates and measure the distribution of the logit vector displacement on a certain data sample. As shown in Figure 6d, in updates between ensembles, the logit vectors are less likely to exhibit significant displacement[5]. This suggests that the ensemble may be reducing the negative flip rate through the reduction of displacements of the logit vectors.

---

[4]"Labrador retriever" (n02099712) and "French bulldog" (n02108915).

[5]Data sample is randomly selected. We present the same visualization on more data points in Appendix E.

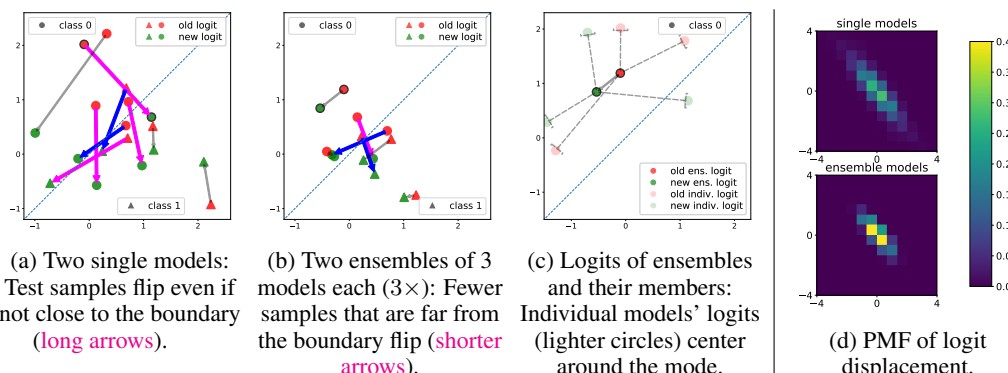

(a) Two single models: Test samples flip even if not close to the boundary (long arrows).

(b) Two ensembles of 3 models each (3×): Fewer samples that are far from the boundary flip (shorter arrows).

(c) Logits of ensembles and their members: Individual models' logits (lighter circles) center around the mode.

(d) PMF of logit displacement.

Figure 6: **Visualization of a 2-class example**. **(a-c)**: Two-class logits of two single models and/or ensembles. ▲ and ● refer to the ground-truth classes, while red and green data points refer to old and new model's logits. Magenta arrow, blue arrow, and gray arrow link negative flip, positive flip, and consistent (either both correct or both wrong) prediction pairs. All dots with black borders are depicting the same image. **(d)**: Estimated probability mass function (PMF) of logit displacement between two single models or ensembles. The $x, y$-axes denote the two classes' logit displacement. The heatmap value denotes the estimated probability density. The ensemble's co-variance is significantly smaller than the single model. The figure is best viewed in color.

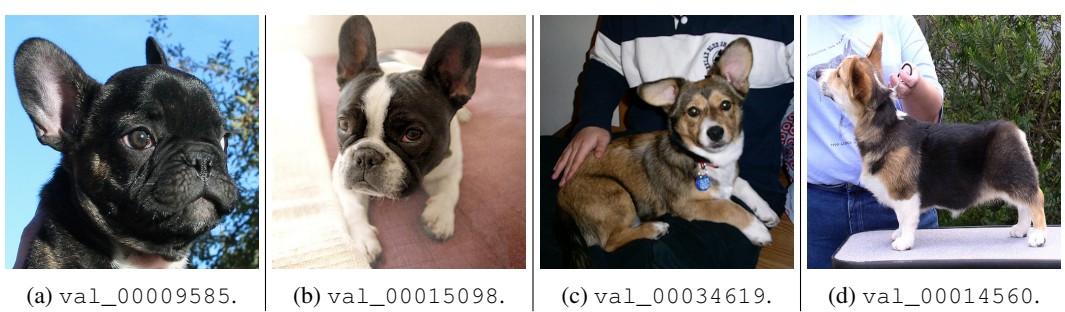

(a) `val_00009585`.  (b) `val_00015098`.  (c) `val_00034619`.  (d) `val_00014560`.

Figure 7: **Four example images for visualization**. The two classes we select here are "French bulldog" (n02108915) and "Welsh Corgi" (n02113023).

# E    VISUALIZATION ON MORE DATA POINTS

As mentioned in Appendix D and Section 3.2, we provide more examples to verify our hypothesis. We select four images of two classes from ImageNet (Deng et al., 2009), which are illustrated in Figure 7, as input data. With these input images, the estimated probability mass function (PMF) of logit displacement between two single model and two ensembles are shown in Figure 8. We can observe that the logit displacements are reduced with ensembles, which verifies our hypothesis of output logit vectors are actually independent and identically distributed (*i.i.d*) random variables and with multi-dimensional central limit theorem (CLT), their sum is a normal distribution (Equation 1).

To verify our hypothesis in higher dimension space, we train a standard ResNet-18 on full ImageNet dataset with 256 random seeds. We take the images in Figure 7 as inputs and illustrate the $\ell_2$ norm histogram of logit displacement between two random ensembles with different ensemble sizes in Figure 9. For heterogeneous case, we train a standard ResNet-50 on full ImageNet dataset and observe the $\ell_2$ norm histogram of logit displacement between a random ResNet-18 ensembles and a random ResNet-50 ensembles with different ensemble sizes. The results are shown in Figure 10.

# F    REPRESENTATION LANDSCAPE OF PC-TRAINING AT THE FEATURE LEVEL

We have discussed the representation landscape of PC-Training in Section 3.1 at the *logit* space and provide some more data points above. The analysis can be done in the *feature* space as well. The main challenge is that features from two arbitrary models are not directly comparable and we address this *feature interoperability* issue by Backward-Compatible Training (Shen et al., 2020).

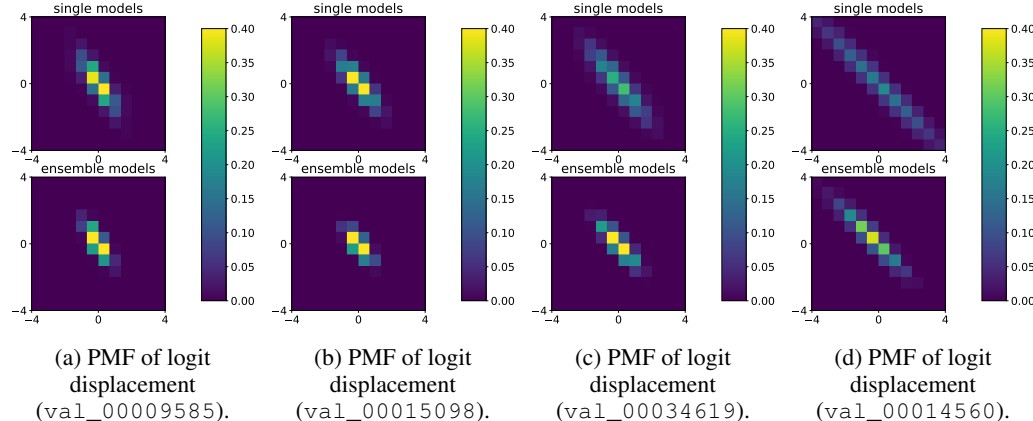

(a) PMF of logit
displacement
(`val_00009585`).

(b) PMF of logit
displacement
(`val_00015098`).

(c) PMF of logit
displacement
(`val_00034619`).

(d) PMF of logit
displacement
(`val_00014560`).

Figure 8: Estimated probability mass function (PMF) of logit displacement between two single models or ensembles. The $x, y$-axes denote the two classes' logit displacement. The heatmap value denotes the estimated probability density. The ensemble's co-variance is significantly smaller than the single model. The figure is best viewed in color.

We first introduce the BCT method and then derive that formulation to attain the feature at the *penultimate layer* of an ensemble. Based on these we can analyze two-dimensional examples and the higher-dimension validation experiments.

**Preliminaries.** Shen et al. (2020) propose an approach termed BCT to align two arbitrary deep models so that the embeddings are interoperable with each other. Formally speaking, a model $\mathcal{M}$ includes an embedding module ($z = \mathcal{F}(x)$, *a.k.a backbone*) and a classification layer ($s = \mathcal{H}(z)$, *a.k.a head*) on top, *i.e* $\mathcal{M}(x) = \phi(x) = (\mathcal{H} \circ \mathcal{F})(x)$. Given a reference model $\mathcal{M}^{(\text{ref})}$, BCT imposes a loss term so that two model heads are close, *i.e* $\mathcal{H}_i^{(\text{bct})} \sim \mathcal{H}^{(\text{ref})}$[6]. As a result, $\mathcal{F}^{(\text{bct})}(x)$ and $\mathcal{F}^{(\text{ref})}(x)$ lie in a same vector space and are thus comparable, *i.e* $\mathcal{F}^{(\text{bct})}(x) \sim \mathcal{F}^{(\text{ref})}(x)$, regardless of the underlying architecture.

**Ensemble of many feature-interoperable models.** It is noteworthy that feature interoperability does **not** affect NFR as reported by Yan et al. (2021). We also re-validate that two models, $\mathcal{F}_1^{(\text{bct})}(x)$ and $\mathcal{F}_2^{(\text{bct})}(x)$, trained using BCT *w.r.t* $\mathcal{M}^{(\text{ref})}$ have similar NFR compared to two without BCT. However, their features are comparable, *i.e* $\mathcal{F}_1^{(\text{bct})}(x) \sim \mathcal{F}_2^{(\text{bct})}(x) \sim \mathcal{F}^{(\text{ref})}(x)$. So is any linear combination in between.

The arguments hold when the number of feature-interoperable models $n$ increases. Therefore, if we write down their averaged logits, we can factor out the head, *i.e*

$$\phi^{(\text{bct,ens})}(x) = \frac{1}{N} \sum_n \phi_n^{(\text{bct})}(x) = \frac{1}{N} \sum_n \left( \mathcal{H}_n^{(\text{bct})} \circ \mathcal{F}_n^{(\text{bct})} \right)(x) \tag{8}$$

$$\approx \frac{1}{N} \sum_n \left( \mathcal{H}^{(\text{ref})} \circ \mathcal{F}_n^{(\text{bct})} \right)(x) = \mathcal{H}^{(\text{ref})} \circ \left( \frac{1}{N} \sum_n \mathcal{F}_n^{(\text{bct})}(x) \right). \tag{9}$$

It implies that the averaged feature can be viewed as this ensemble's feature, *i.e* $\mathcal{F}^{(\text{bct,ens})}(x) = \frac{1}{N} \sum \mathcal{F}_n^{(\text{bct})}(x)$.

**A two-dimensional example.** Similar to Section D, to illustrate the behavior of negative flips in the feature space, we create a toy example by selecting three classes[7] from ImageNet (Deng et al., 2009) and training a ResNet-18-like models with a slight modification such that the penultimate layer's dimension is changed to 2. The feature level visualization is presented in Figure 11a and 11d. We

---

[6]In fact if we assume that $\mathcal{H}^{(\text{bct})}$ and $\mathcal{H}^{(\text{ref})}$ have the same shape, we can also do as follows: we train $\mathcal{M}^{(\text{ref})}$ and then $\mathcal{M}^{(\text{bct})}$ with parameters randomly initialized except the head copies weights from $\mathcal{H}^{(\text{ref})}$ and is *fixed*. Nevertheless, we follow BCT's formulation since it is more generic.

[7]"Labrador retriever" (n02099712), "Weimaraner" (n02092339), and "French bulldog" (n02108915).

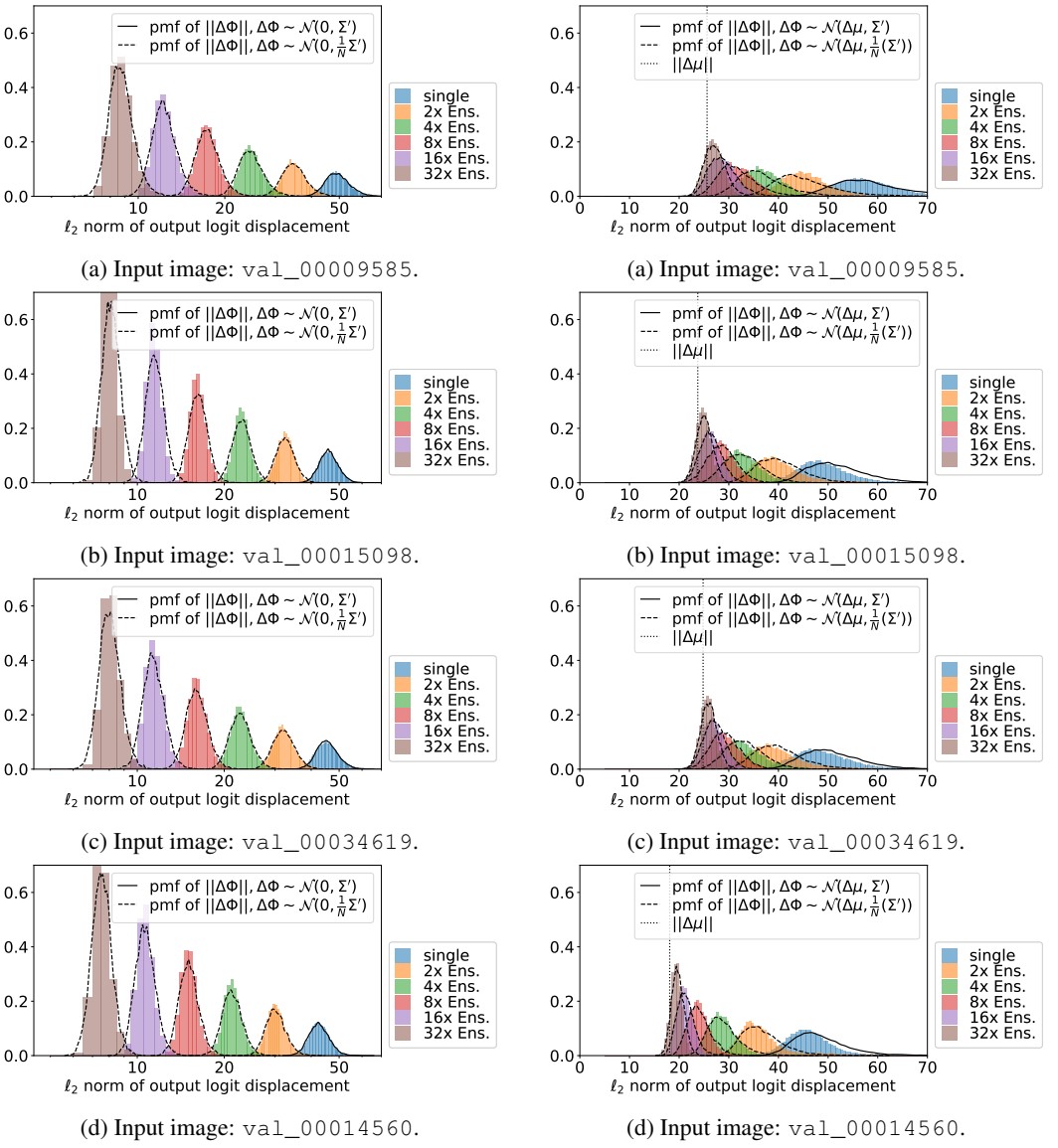

Figure 9: $\ell_2$ **norm histogram of logit displacement between two random ensembles.** The bin size is 0.5. Two random ensembles are the same type (homogeneous ResNet-18 *vs* homogeneous ResNet-18). $\Delta\boldsymbol{\mu} = \boldsymbol{\mu}_1 - \boldsymbol{\mu}_2 = \mathbf{0}, \boldsymbol{\Sigma}' = 2\boldsymbol{\Sigma}_1 = 2\boldsymbol{\Sigma}_2$.

Figure 10: $\ell_2$ **norm histogram of logit displacement between two random ensembles.** The bin size is 0.5. Two random ensembles are different types (homogeneous ResNet-18 vs homogeneous ResNet-50). $\Delta\boldsymbol{\mu} = \boldsymbol{\mu}_1 - \boldsymbol{\mu}_2 \neq \mathbf{0}, \boldsymbol{\Sigma}' = \boldsymbol{\Sigma}_1 + \boldsymbol{\Sigma}_2$.

can observe the similar observations as in the logit space after the penultimate layer features are aligned with BCT (Shen et al., 2020).

**Validations on higher dimensions.** We repeat the high-dimensional validation in text on the penultimate layer features, the results are shown in Figure 12. We see that the PMF curve fits the histogram of single models well, implying that feature of these models could indeed follow a Normal distribution. We conduct the same experiments above on many more images and the conclusion holds well. If we move to ensembles of $m$ models each, the feature difference follows another normal distribution whose co-variance matrix is scaled by a factor of $m$, *i.e* $\Delta\boldsymbol{z}^{(\text{ens})} \sim \mathcal{N}\left(\mathbf{0}, \frac{2}{m}\boldsymbol{\Sigma}\right)$. We demonstrate that the rest of histograms are indeed consistent with the estimated PMF of $\|\Delta\boldsymbol{z}^{(\text{ens})}\|^2$ (dashed lines in Figure 12).

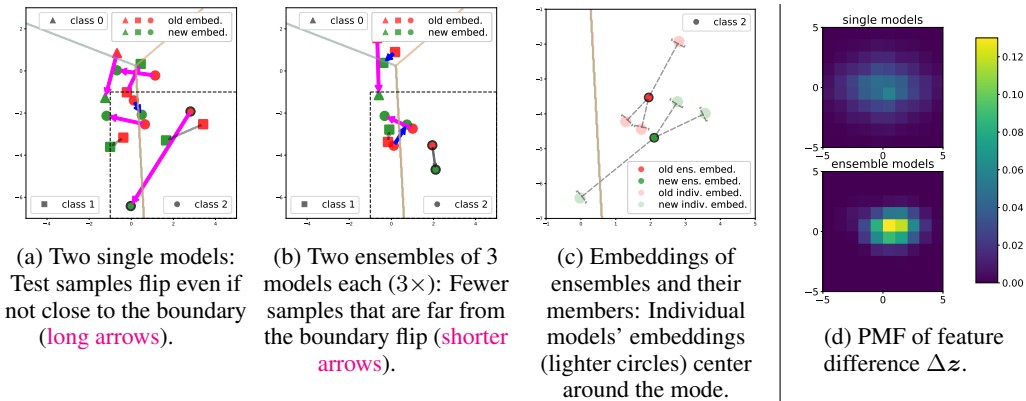

(a) Two single models: Test samples flip even if not close to the boundary (long arrows).

(b) Two ensembles of 3 models each (3×): Fewer samples that are far from the boundary flip (shorter arrows).

(c) Embeddings of ensembles and their members: Individual models' embeddings (lighter circles) center around the mode.

(d) PMF of feature difference $\Delta z$.

Figure 11: **Visualization of a 3-class 2-dimensional example**. **(a-c)**: 2D feature embedding of two single models or ensembles. ▲, ■, and ● refer to the ground-truth classes for each sample. Red and green data points refer to old and new model's embeddings. Magenta arrow, blue arrow, gray arrow link negative flip, positive flip, and consistent (either both correct or both wrong) prediction pairs. All dots with black borders are depicting the same image. **(d)**: Estimated probability mass function (PMF) of feature difference between two single models or ensembles. The $x$- and $y$-axes denote the 2D feature difference. The heatmap value denotes the estimated probability density. The ensemble's co-variance is significantly smaller than the single model. The figure is best viewed in color.

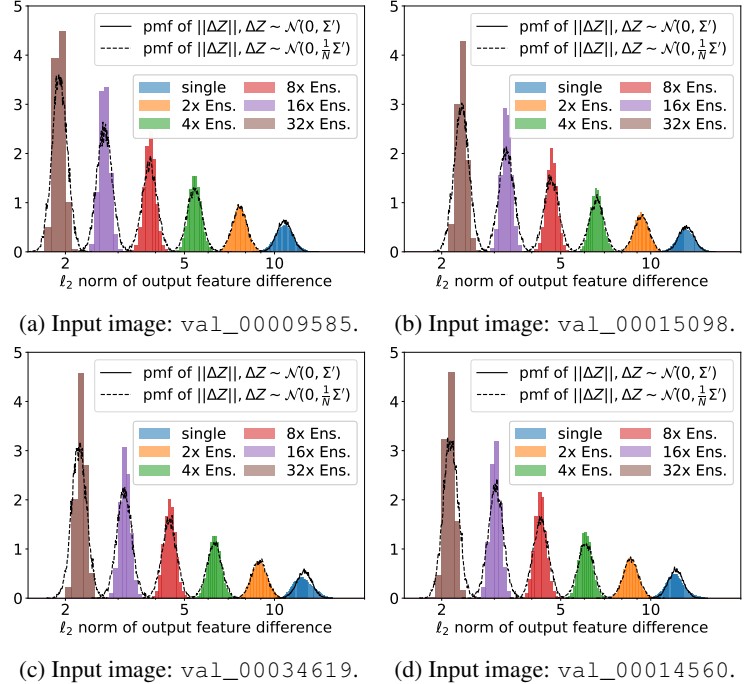

(a) Input image: `val_00009585`.

(b) Input image: `val_00015098`.

(c) Input image: `val_00034619`.

(d) Input image: `val_00014560`.

Figure 12: $\ell_2$ **norm histogram of feature difference between two random ensembles.** Note that the bin size is 0.1. Two random ensembles are of the same type (ResNet-18 *vs* ResNet-18). $\Delta\boldsymbol{\mu} = \boldsymbol{\mu}_1 - \boldsymbol{\mu}_2 \neq \mathbf{0}, \boldsymbol{\Sigma}' = \boldsymbol{\Sigma}_1 + \boldsymbol{\Sigma}_2$. We also plot the simulated probability mass function (PMF): the solid line for the norm of a simulated normal distribution $\mathcal{N}\left(\Delta\boldsymbol{\mu}, (\boldsymbol{\Sigma}_1 + \boldsymbol{\Sigma}_2)\right)$ whose parameters are estimated from all available single models; the dashed lines for extrapolated distribution $\mathcal{N}\left(\Delta\boldsymbol{\mu}, \frac{1}{m}(\boldsymbol{\Sigma}_1 + \boldsymbol{\Sigma}_2)\right)$. Consistency between the ensembles' histograms and PMFs supports our hypotheses in Section F.

