# OpenReview forum: "ELODI: Ensemble Logit Difference Inhibition for Positive-Congruent Training"
_ICLR.cc/2023/Conference — Submitted to ICLR 2023_

### Official Review · Reviewer_uaPa · 2022-10-25

**Confidence:** 4
**Correctness:** 3
**Technical Novelty And Significance:** 3
**Empirical Novelty And Significance:** 4
**Recommendation:** 6

**Clarity, Quality, Novelty And Reproducibility:**

Clarity:
The paper is well written, organized, and technically detailed. The used notation is clear and consistent throughout the paper.

Quality:
The design and justifications for the proposed Ensemble Logit Difference Inhibition method are technically sound. The observations made regarding the reduction in NFR and inference cost introduced by the proposed method as well as its capability to maintain image classification accuracy gains are empirically well supported. Overall, the work seems to be well developed and mature in terms of quality.

Novelty:
From a methodological perspective, the contribution of this work can be considered incremental since it is built upon the well-established knowledge distillation framework that utilizes the proposed simple yet well-crafted Logit Difference Inhibition (LDI) objective. Nevertheless, the authors provide an extensive analysis on the role of deep ensembles in a knowledge distillation scheme which sheds light on their capability of reducing NFR and obtaining performance for PC-training with a single model, at the running cost of a single model. To the best of my knowledge, this is the first work to conduct such an extensive analysis regarding the aforementioned points.

Reproducibility:
The experiments were conducted on widely-used image classification benchmark datasets which are publicly available. On the other hand, the code for the proposed ELODI method is not made available by the authors in the present anonymized version, however, the architecture of the method is clearly explained in the paper, thus it is safe to say that one should be able to implement ELODI by following its detailed description in Section 4.

**Details Of Ethics Concerns:**

Not applicable.

**Strength And Weaknesses:**

Strengths:

* ELODI trains a deep homogeneous ensemble that can achieve Positive-Congruent Training (PC-Training) performance and be distilled to a single model (which does not necessarily need to be a specific legacy model). This allows the proposed ELODI method to reduce NFR while retaining the accuracy gain introduced by the new model.

* The authors introduce Logit Difference Inhibition (LDI), a knowledge distillation objective that penalizes significant differences between the logits of the reference homogeneous ensemble and the single student model. Such an objective allows for effective distillation of the reference ensemble to a single student model.

* ELODI exhibits efficient inference as it does not require the homogeneous reference ensemble to be evaluated at inference time but rather only the single student model.

* The choice of homogeneous ensembles as reference ensembles is empirically well supported by the findings summarized in Figure 4. Moreover, the hypothesis of reducing logit displacement magnitude (on which the introduced LDI objective relies) is validated empirically in Section 3.2.

* The authors have conducted extensive experiments on multiple image classification benchmarks to assess the effectiveness and efficiency of ELODI in comparison with several PC-training and ensemble methods. The experimental results demonstrate that ELODI performs positive congruent training by reducing NFR with large logit displacement and reducing the variance of the ensemble’s logits. It is also empirically demonstrated that this behavior can be transferred to a single model which considerably reduces the inference time. For instance, this allows ELODI to achieve comparable ER-NFR performance to that of the ensemble paragon [Yan et al., 2021] at the inference cost of a single model.

* The authors have conducted a plethora of additional experimental studies to assess the effect of various aspects including:

    (1) model update settings (including fine-tuning on other datasets, transitivity of NFR reduction induced in chain of model updates, and ELODI’s applicability to updates across dissimilar architectures);

    (2) design choices (including the effects of using homogeneous instead of heterogeneous ensembles for guidance, changes in the guiding ensemble’s architecture, and different choices of distillation loss functions);

    (3) exploratory parameter analysis assessing the effects of the loss weight $\alpha$, the size of the reference ensemble as well as the manner in which the logits are inferred (offline compared to online).

-------------------------------------------------------------------------------------------------

Weaknesses:

* At the end of the introductory paragraph of Section 3, the authors claim that “the averaging operation in deep ensembles is also often performed in the logit space (Lakshminarayanan et al., 2017), which makes it interesting for our study.” Nevertheless, I am not confident how often this is the case, namely because in classification settings voting schemes are typically used upon applying the activation function of choice (e.g., softmax) to the logits estimated by each member model rather than averaging the logits. I would encourage the authors to further clarify their motivation behind the choice of the averaging aggregation scheme in the logit space, or consider revising the aforementioned statement in the paper.

* The assumption that the logits outputted by the base members of a homogeneous ensemble follow a normal distribution is a fair assumption to make. This is also empirically supported in Section 3.2. However, [1] suggests that, for this to hold, the base networks need to be trained in a sufficient number of epochs. Therefore, including a brief discussion in Section 3.2 on the relationship between the number of training epochs and the members’ logit distributions would be quite insightful. In addition, I believe that [1] is relevant to this paperr as it discusses logit distributions of deep neural networks and thus I would suggest that the authors consider including it as a part of their related work.

    [1] Seguin, L., Ndirango, A., Mishra, N., Chung, S., & Lee, T. (2021). Understanding the Logit Distributions of Adversarially-Trained Deep Neural Networks. arXiv preprint arXiv:2108.12001.

* In Section 3.2, while discussing Figure 3a, the authors state the following: “If we move to the ensemble case, the logit displacement follows another normal distribution with scaled co-variance matrix”. However, according to Figure 3a, apart from the scaled co-variance matrix, the mean is shifted. This should also be included in the same discussion.

-------------------------------------------------------------------------------------------------

Minor weaknesses:
There are also certain grammatical and typographical errors and remarks that require attention. Some of them are summarized as follows:
- In the “Contributions” paragraph on page 2, “the” should be removed from the phrase “obtain the their performance for PC-training”. Later in the same paragraph, the hyphen should be removed from “with re- spect to”.
- Section 3.1, at the end of page 3: In the phrase “Each $\phi(x)$ or $\phi(x)$”, I believe that one of the $\phi(x)$’s should be corrected.
- In the second paragraph of Section 4.1, replace “an model pair” with “a model pair”.
- Section 4.2: In the sentence right after Eq. (6), it is mentioned that “difference below $\xi$ is tolerated”; however, according to Eq. (6) it seems that a difference below $\xi$ is truncated, i.e. not tolerated. Later in that same paragraph, “when number of classes are extremely largencated” should be corrected to “when the number of classes is extremely large”.


**Summary Of The Paper:**

In this work, the authors repurpose the well-known knowledge distillation paradigm, which is typically used for improving model accuracy, for the task of transferring the capability of reducing negative flip rate (NFR) from an ensemble to a single model. To this end, the authors devise a method called Ensemble Logit Difference Inhibition (ELODI). ELODI leverages the advantages of deep ensembles for NFR reduction at the running cost of a single model by (1) training deep networks using a Logit Difference Inhibition (LDI) loss with respect to a reference ensemble and then (2) using the resulting single model to perform inference.

**Summary Of The Review:**

This paper leveraged deep ensembles for NFR reduction and showed that they can be distilled into single models by means of a Logit Difference Inhibition (LDI) objective. Despite the methodologically incremental contribution, the proposed ensemble method is capable of reducing NFR while maintaining accuracy and can be distilled to a readily deployable single model which exhibits much lower inference time compared to the deep ensemble. The latter in particular makes this work applicable to large-scale real-world systems where online inference and low latency are paramount. Overall, although there are certain weak points that need to be addressed (listed in the “Weaknesses” part of this review), the strengths of this work certainly outweigh its weaknesses. Therefore, I consider this paper to be a fairly good fit for ICLR and I recommend that it is accepted.

---

> ### Author Response · Authors · 2022-11-18
> **Response to Reviewer uaPa**
>
> Thank you for your comments and feedbacks! We address your concerns below.
>
> ----
> Q1. Clarify their motivation behind the choice of the averaging aggregation scheme in the logit space.
>
> A1. We agree with the reviewer that there is a line of work where the voting scheme is used upon applying a softmax to the logits, for example, [C1,C2]. However, recent works [C3,C4] also conduct model ensembling by averaging logits and achieve impressive improvement. We have included these references to support our practice. Next, we would like to argue that the logit space also has the following desirable property for our study:  **Without** the softmax operation that squashes the score and skews the distribution, the raw output of logits now follows a normal distribution, which is supported in Fig 6(d) and Fig 8. We have added the discussion in the revised text.
> Finally, we add an additional experiment of Elodi using probability scores after softmax. We can see that averaging on the normalized probabilities cannot reduce NFR. We also tried reducing the "temperature" (t) as in focal distillation but didn't see much gain.
>
> | method | ResNet18 ER |  ResNet50 ER  | NFR |
> | ---- | ---- |  ----  | ---- |
> | non-treatment baseline	 | 30.24 | 24.66 | 4.30 |
> |  Elodi  | 31.34 | 23.15 | 2.18 |
> |  w/ softmax (1/t=1) | 30.15 | 25.57 | 5.11 |
> |  w/ softmax (1/t=20) | 29.99 | 24.52 | 4.17 |
>
> [C1] Simonyan, K., & Zisserman, A. (2015). Very deep convolutional networks for large-scale image recognition. ICLR.
>
> [C2] Lakshminarayanan, B., Pritzel, A., & Blundell, C. (2017). Simple and scalable predictive uncertainty estimation using deep ensembles. NeurIPS.
>
> [C3] Gontijo-Lopes, R., Dauphin, Y., & Cubuk, E. D. (2022). No one representation to rule them all: Overlapping features of training methods. ICLR.
>
> [C4] Wortsman, M., Ilharco, G., Gadre, S. Y., Roelofs, R., Gontijo-Lopes, R., Morcos, A. S., Namkoong, H., Farhadi, A., Carmon, Y., Kornblith, S., & Schmidt, L. (2022). Model soups: averaging weights of multiple fine-tuned models improves accuracy without increasing inference time. ICML.
>
>
> ----
> Q2. Discussing relationship between the number of training epochs and the members’ logit distributions [C5].
>
> A2. Thanks for bringing up this important factor in supporting the assumption. In fact, we follow the standard training recipe (Appendix 1) where the number of epochs for training (90 epochs) should be sufficient for the assumption of [C5] to hold. We have also highlighted this in the updated text. Please refer to the first paragraph of Sec 3.2.
>
> [C5] Seguin, Landan, et al. Understanding the Logit Distributions of Adversarially-Trained Deep Neural Networks. arXiv preprint arXiv:2108.12001.
>
> Q3. There are also certain grammatical and typographical errors and remarks that require attention.
>
> A3. Thank you for the careful proofreading. We have fixed all these grammatical and typographical errors in the latest version uploaded to OpenReview.

---

> > ### Comment · Reviewer_uaPa · 2022-11-20
> > **Follow-up on authors' response**
> >
> > I would like to thank the authors for their point-by-point response to my review. The authors have clarified my major concerns related to (1) the choice of the averaging aggregation scheme in the logit space, and (2) the relationship between the number of training epochs and logit distributions of the individual ensemble members; and made the appropriate revisions in the paper. Therefore, I consider that addressing the aforementioned two points further supports my initial recommendation for this paper.

---

> > > ### Author Response · Authors · 2022-11-30
> > > **Reply**
> > >
> > > Thank you again for your time in reviewing our paper and appreciating the value of our work!

---

> ### Author Response · Authors · 2022-12-10
> **Follow-up on your recent rating change**
>
> Dear Reviewer uaPa,
>
> We noticed that there is a recent change in your rating. As you commented earlier that our responses have supported your initial rating, which was “8-good paper, accept”. Please kindly let us know whether you have other concerns not addressed by our responses. Thank you.
>
> Best, Authors of Paper #8

---

> > ### Comment · Reviewer_uaPa · 2022-12-12
> > **Re: Follow-up on your recent rating change**
> >
> > Dear Authors,
> >
> > Correct, your response did support my initial recommendation, nevertheless, kindly note that the recent change in the recommendation grade is a result of an ongoing discussion between the AC and the other reviewers which is a separate discussion that was initiated after the period of discussion with the authors.
> >
> > Best,
> >
> > Reviewer uaPa

---

### Official Review · Reviewer_V2YQ · 2022-10-25

**Confidence:** 3
**Correctness:** 4
**Technical Novelty And Significance:** 3
**Empirical Novelty And Significance:** 3
**Recommendation:** 6

**Clarity, Quality, Novelty And Reproducibility:**

- The paper is well-written and easy to follow.
- There are minor typos throughout the text (e.g., "res- pect").
- Figure 1 is in page 2, but is referenced for the first time in page 5.
- Figures 3 and 4 are somewhat hard to understand, since there is too much
  content packed into very little space.
- Algorithm 1 seems unnecessary, given that the procedure is explained
  throughout the text.
- The novelty of the paper is the distillation loss function targeted at
  approximating the logits between a new model and an ensemble. The authors
  compare the proposed loss with several baselines and conclude that it gives
  better results.
- The paper seems to be easy to reproduce (hopefully the authors will publish
  their source code).

**Strength And Weaknesses:**

*Strengths:

- The paper is well-motivated, since the problem the authors present frequently
  occurs in production systems.
- The problem of negative flips in the context of ensembles is clearly
  explained in Section 3 and further details the experiments in (Yan et al.
  2021).
- The experimental section compares the results of ELODI with several other
  state-of-the-art techniques.

*Weaknesses:

- ELODI works mainly for homogeneous ensembles. This goes against one of the
  biggest advantages of using ensembles, which is to combine models with
  different architectures.
- Training a model using ELODI requires one to first train an ensemble and, as
  the authors themselves point out, the number of models in the ensemble
  drastically impacts the quality of the distilled model.

**Summary Of The Paper:**

The authors tackle the problem of updating existing models in production-scale
systems while performing Positive-Congruent Training (minimize negative flip
rate and error rate simultaneously). To do so, they make two key observations:

- Previous studies have demonstrated that ensembles can reduce NFR without
  negative accuracy impact.
- Using ensembles may not be a viable strategy in real applications due to
  computational costs.

Based on these observations, the authors propose ELODI, a novel model
distillation method that seeks to achieve PC-Training performance of ensembles
using a single model. First, they analyze how the distribution of logit spaces
of both homogeneous and heterogenous ensembles behave and conclude that, as
ensembles become larger, NFR consistently decreases. This is further validated
with experiments. Next, they propose a novel, margin-based distillation loss
function, namely LDI, that attempts to mimic the variance reduction phenomenon
present in ensembles. The authors perform extensive experiments and conclude
that ELODI has similar performance (NFR and ER) to an ensemble method with
reduced inference costs.

**Summary Of The Review:**

The authors propose ELODI, a distillation technique that reaps both the
benefits of ensembles to reduce NFR and ER and distillation to mimic the
behavior of the said ensemble using a single model. Although mainly limited to
homogeneous ensembles, ELODI has the potential to positively affect deep
learning model deployment in production.

---

> ### Author Response · Authors · 2022-11-18
> **Response to Reviewer V2YQ**
>
> Thank you for your comments and feedbacks! We address your concerns below.
>
> ----
> Q1. ELODI works mainly for homogeneous ensembles. This goes against one of the biggest advantages of using ensembles, which is to combine models with different architectures.
>
> A1. In Table 4, we have compared the effectiveness of using heterogeneous ensemble and homogeneous ensemble in ELODI. We also summarized the key numbers here (where “non-treatment baseline”,  “heterogeneous ens.” and “homogeneous ens.” refers to the first, fifth, and eighth row in Table 4.
>
> |  method  | ResNet18 ER | ResNet50 ER | NFR |
> | -------------------------- | ------- | ------ | ----- |
> | non-treatment baseline | 30.24 | 24.66 | 4.30 |
> | heterogeneous ens. | 32.73 | 23.14 | 2.25 |
> | homogeneous ens. | 31.32 | 23.15 | 2.18 |
>
> The results clearly show that using a homogeneous ensemble leads to slightly better reduction of NFs, which corresponds well with our analysis in Sec 3. Additionally, using homogenous ensembles is easier to implement in practice because: 1) There is no need to design a set of different architecture when training a new model; (2) the good practices identified in training the target model from scratch can be applied in ELODI.
>
> ----
>
> Q2. Training a model using ELODI requires one to first train an ensemble and, as the authors themselves point out, the number of models in the ensemble drastically impacts the quality of the distilled model.
>
> A2. We study the effect of the ensemble size on NFR in Fig. 5. We can see that NFR quickly decreases when increasing the ensemble size from 1 to 4 and then plateaus. Using an ensemble with 4 or 8 models is generally feasible in practice.
>
> ----
>
> Q3. Figure 1 is on page 2, but is referenced for the first time on page 5.
>
> A3. Thank you for the suggestions. This is a very good point! We've changed the draft so that Figure 1 is introduced in the Sec 1 right below itself. Please refer to the latest version on OpenReview.

---

> > ### Comment · Reviewer_V2YQ · 2022-11-29
> > **Response to authors**
> >
> > Thank you for your response. I keep my belief that this is an interesting work, above the acceptance threshold.

---

> > > ### Author Response · Authors · 2022-11-30
> > > **Reply**
> > >
> > > Thank you again for your time in reviewing our paper and appreciating the value of our work!

---

### Official Review · Reviewer_38Tg · 2022-10-26

**Confidence:** 2
**Correctness:** 3
**Technical Novelty And Significance:** 2
**Empirical Novelty And Significance:** 2
**Recommendation:** 5

**Clarity, Quality, Novelty And Reproducibility:**

Overall the paper is written clearly. I feel the novelty of the approach is a bit limited since the key part is on modifying the loss function used in knowledge distillation; the part on the distribution of ensemble distributions is only empirically without any theoretical supports.

Some minor things:

- Figure 2 the blue color is not accurate when laid over by the orange.
- Better to write down exact formulation of L_CE in page 6.

**Strength And Weaknesses:**

Strength

- The analysis on the properties of the distribution of ensemble predictions is interesting, and the simulation studies are clear and supportive of the findings.
- The algorithm is well presented with detailed follow up discussions.

Weakness
- I am not super clear on the motivation of the proposed method. The loss function of the distillation algorithm is a key part but it's basically just cross entropy plus a term to panelize l_2 difference in the logit. Despite being used for model updating, are there any other benefits can be discussed, such as smoothing the landscape of the student model? It seems to me using and additional LDI term could have some alternative motivations, and the use case seems a bit unnatural to me.
- Although the discussion on the properties of ensemble prediction distributions are empirically correct, can the authors provide more discussions in this area? For example, where is the randomness coming from? Initialization, randomness in training algorithms, etc? For trees, there have been some statistical results (see for example https://arxiv.org/pdf/1510.04342.pdf, https://www.jmlr.org/papers/volume17/14-168/14-168.pdf?ref=https://githubhelp.com). I wonder whether some primitive results can be discussed here.

**Summary Of The Paper:**

This paper proposes a new method in the area of Positive-Congruent Training, by distilling a homogeneous ensemble to a single student model. The authors carefully analyze the properties of logic displacement magnitude in ensembles, and present a method called Ensemble Logit Difference Inhibition (ELODI). The proposed approach is validated on two standard image classification datasets.

**Summary Of The Review:**

This paper introduces a new knowledge distillation method with the goal to reducing NFR. Although the analysis and the methods are well presented, I feel the novelty of the approach is a bit limited since the key part is on modifying the loss function used in knowledge distillation; and the part on the distribution of ensemble distributions is only empirically without any theoretical supports.

That being said, I think if the authors can provide a better explanation on the benefits of the proposed loss function, and provide more theoretical discussion on the prediction distributions of ensembles, the paper can be much improved.

I mark the paper as marginally below the threshold, but also note I am not familiar with this specific application area. So I am willing for further discussions.

---

> ### Author Response · Authors · 2022-11-18
> **Response to Reviewer 38Tg**
>
> Thank you for your comments and feedbacks! We address your concerns as follows.
>
> ----
> Q1. The motivation of the proposed method.
>
> A1. Our main goal is to have a single model that resembles the behavior of a model ensemble, which has a reduced NFR compared to single models without any treatment. We achieve this goal by repurposing knowledge distillation. Specifically, the proposed LDI term inhibits significant difference of logit vectors’ elements between the distilled model and the ensemble instead of pursuing exact match. We empirically show that a series of models trained from this objective achieves a significantly reduced NFR. We also observe that the proposed LDI loss can also improve the single model's accuracy because of its inherent distillation formulation.
>
> ----
> Q2. Where is the randomness coming from? More discussion on this area.
>
> A2. The randomness can be categorized as follows: (1) the initialization of model parameters, (2) the order of samples in each batch of SGD, and (3) data augmentation (e.g. RandomResizedCrop and RandomHorizontalFlip). Low-level acceleration libraries such as cuDNN can also include nondeterminism. We empirically observed that any one kind of change will lead to a noticeable amount of negative flip samples. Some recent works have also shown empirical results of different kinds of randomness or nondeterminism [C1,C2,C3]. Theoretical landscape analysis of deep ensembles is a very interesting and important direction to explore, but is beyond the scope of our paper.
>
> [C1] Summers, C., & Dinneen, M. J. (2021). Nondeterminism and instability in neural network optimization. ICML.
>
> [C2] Nagarajan, P., Warnell, G., & Stone, P. (2018). The impact of nondeterminism on reproducibility in deep reinforcement learning. ICML Workshop.
>
> [C3] Madhyastha, P., & Jain, R. (2019). On model stability as a function of random seed. CoNNL.

---

> > ### Comment · Reviewer_38Tg · 2022-11-21
> > **Thanks for the response.**
> >
> > Thanks for providing the explanation to my raised concerns. Agree on Q2 that a more deep understanding is out of the scope.

---

> > > ### Author Response · Authors · 2022-11-30
> > > **Reply**
> > >
> > > Thank you again for your time in reviewing our paper. We hope that we have addressed your concerns with our response. Please do not hesitate to let us know if there are any outstanding questions or comments. We would also kindly ask you to consider your evaluation in light of our response.

---

### Official Review · Reviewer_asve · 2022-11-04

**Confidence:** 4
**Correctness:** 3
**Technical Novelty And Significance:** 4
**Empirical Novelty And Significance:** 4
**Recommendation:** 5

**Clarity, Quality, Novelty And Reproducibility:**

The paper is sufficiently novel and reproducible, and also mostly clear, except for the shortcomings highlighted above.

**Strength And Weaknesses:**

Strengths:
* Theoretical analysis of the differences of average logits in ensembles;
* The paper is mostly clearly written, with plenty of insightful figures and tables;
* The proposed method ELODI is effective in reducing NFR while mostly retaining accuracy.

Weaknesses:
* At the end of Section 4, it is stated that the logit matching loss (Hinton et al., 2015) is a special case of the proposed LDI term. Therefore, I find it absolutely necessary to do an ablation study to see whether the proposed method improves over using the plain logit matching loss. That is, the paper should have experimental comparisons of $\xi=0.1$ against $\xi=0$. Currently there is no such comparison in the paper (nor in the appendices).
* The theoretical part first starts by making claims about any particular fixed $x$, while considering the training process (or training seed) as a source of randomness, hence the random variable $\phi^{(i)}(x)$. This nicely gives results stated by Eqs. (1), (2) and (3). However, in Section 3.2 there seems to be a shift to consider $x$ also as a random quantity, because there is a single Gaussian distribution about the whole architecture, e.g. see the last paragraph of page 4 with $\phi\sim\mathcal{N}(\mu_1,\Sigma_1)$. This is confusing and should be clarified. If there is now indeed a single Gaussian per architecture, what is the theoretical justification behind that? If there is still only a Gaussian per instance, then how are the covariance matrices estimated for Figure 3?
* The first variance term $Var(\hat{\phi}(x))$ in Eq.(5) does not make sense to me. As minimization is over $\hat{\phi}(x)$ and thus this quantity is bound within the expression to be minimized, then what is the random variable that the variance operator is applied to?

Minor weaknesses:
* From the introduction it is not quite clear whether the term positive-congruent training is novel or not;
* The text about Figure 2 suggests that particularly the negative flips with large cross-model logit displacement are reduced by ensembles, highlighting the percentage of 90.7% in the red box. However, it would then also be important to provide the respective percentage outside the red box. Is it significantly smaller?
* The last rows of page 3 have '$\phi^{(i)}$ and $\phi^{(i)}$' instead of '$\phi^{(i)}$ and $\psi^{(i)}$';
* Sometimes $\phi$ is in bold and sometimes not (e.g. in Eqs (1) and (3)), not being consistent.
* The number of current Figure 6 should be Figure 5 instead. The text already refers to it as Figure 5.
* In Figure 6b (that is 5b), it has not been explained what ensemble size 0 means.
* It has not explained clearly enough what the difference between online and offline distillation is.
* The current theory and the discussion around it say nothing about decreasing the difference between $\mu_1$ and $\mu_2$ which could also be a separate goal. In principle, this might possibly be achieved by changing the training objective for the members of the ensemble. Maybe it is a bad idea, but I feel this should be at least mentioned.

Suggestions:
* The acronyms LDI and ELODI are both about logit difference inhibition: wouldn't it be better to use the same suffix, such as LDI and ELDI, or LODI and ELODI?

**Summary Of The Paper:**

The paper addresses the task of updating a model by learning a stronger model while ensuring that formerly correctly classified instances will get wrong predictions as seldom as possible, a quantity called a negative flip rate (NFR). The paper starts with a theoretical analysis of using ensembles in this context. It then proposes a new training objective to distill a single model based on an ensemble, with the goal of minimizing NFR. Experiments on image datasets ImageNet and iNaturalist show the effectiveness of ELODI compared to other NFR-reducing methods.

**Summary Of The Review:**

The paper addresses an important topic and proposes new effective methods. It also studies the theoretical justification behind the proposed methods. In my opinion, an important aspect has not been covered as an ablation study, justifying the difference of the proposed method from the existing logit matching loss by Hinton et al 2015. There are some shortcomings in the theoretical part, as highlighted above in the list of weaknesses.

---

> ### Author Response · Authors · 2022-11-18
> **Part 1 of Response to Reviewer asve**
>
> Thank you for your comments and feedbacks! We address your concerns as follows.
>
> ----
> Q1. Ablation studies on the proposed LDI term.
>
> A1. (1) we empirically select $\xi=0.1$ based on the trade-off shown in the table below; (2) The fact that only penalizing large displacement works equally well or better in NFR reduction guides us to further explore Top-K LDI and reach the generic loss form in Eq (6) for NFR reduction. Top-K LDI has lower compute cost for large classifier cases.
>
> | $\xi $ | R18 ER | R50 ER | NFR |
> | -- | -- | -- | -- |
> | 0.0 |  31.34 | 23.15 | 2.18 |
> | 0.01 |  31.28 | 23.58 | 2.32 |
> | 0.05 |  31.02 | 23.48 | 2.42 |
> | 0.1 |  31.00 | 23.28 | *2.26* |
> | 0.2 | 30.96 | 23.21 | 2.43 |
>
> ----
> Q2. Clarification on the instance-specific logit distribution and how we estimate the covariance matrices.
>
> A2. The analysis in the third paragraph of Section 3.2 is also about a particular fixed image (See "for an arbitrary image" in the first paragraph of Section 3.2). We used $\boldsymbol\phi\sim\mathcal{N}(\boldsymbol\mu_1,\boldsymbol\Sigma_1)$ for brevity but we agree that $\boldsymbol\phi(x)$ is a more clear notion to emphasize Gaussian per instance. Therefore we have changed it accordingly to disambiguate it. We also would like to refer the reviewer to Figure 10 and Figure 11 in the Appendix, which shows such instance-specific distribution varies across different images nontrivially. We've noted this in the revised version.
>
>   To estimate the covariance matrices, we first train one model multiple times with different random seeds and then treat the logits of the same input $ x $ by different models as the observations. Given these observations, we can estimate the mean and covariance matrix for $\boldsymbol\phi(x)$. Specifically, to get Fig 3, we train 256 ResNet-18 and 128 ResNet-50 models with different seeds. Next we randomly pick an image $ x $ from the ImageNet-1k val set and obtain the logits of all models, namely $\boldsymbol\phi^{(1)}(x), \boldsymbol\phi^{(2)}(x), ...\cdots, \boldsymbol\phi^{(256)}(x)$ and $\boldsymbol\psi^{(1)}(x), \boldsymbol\psi^{(2)}(x), ...\cdots, \boldsymbol\psi^{(128)}(x)$. Therefore the estimated means are $\boldsymbol\mu_1(x) = \frac{1}{256}\sum_{i=1}^{256}(\boldsymbol\phi^{(i)}(x))$ and $\boldsymbol\mu_2(x) = \frac{1}{128}\sum_{i=1}^{128}(\boldsymbol\psi^{(i)}(x))$; the estimated covariance matrices are $\boldsymbol\Sigma_1(x)=E[(\boldsymbol\phi(x)-\boldsymbol\mu_1)(\boldsymbol\phi(x)-\boldsymbol\mu_1)^\top]=\frac{1}{256-1}\sum_{i=1}^{256}(\boldsymbol\phi^{(i)}-\boldsymbol\mu_1(x))((\boldsymbol\phi^{(i)}-\boldsymbol\mu_1(x)))^\top$ and $\boldsymbol\Sigma_2(x)=E[(\boldsymbol\psi(x)-\boldsymbol\mu_2)(\boldsymbol\psi(x)-\boldsymbol\mu_2)^\top]=\frac{1}{128-1}\sum_{i=1}^{128}(\boldsymbol\psi^{(i)}-\boldsymbol\mu_2(x))((\boldsymbol\psi^{(i)}-\boldsymbol\mu_2(x)))^\top$.
>
> ----
> Q3. Formulation of Equation (5).
>
> A3. Thanks for spotting this issue. The optimization process is with respect to the model $\phi$ with its parameters. The result of the optimization is then $\hat\phi$. We have updated Equation (4), (5) and (6) in the latest version uploaded to OpenReview.
>
> ----
> Q4. The percentage of negative flips reduced by ensembles inside and outside the red box in Fig 2.
>
> A4. Outside the red box, 80.9% negative flips are reverted, which is significantly lower than 90.7% inside the red box.
>
> ----
> Q5. The last rows of page 3 have '$\phi^{(i)}$ and $\phi^{(i)}$' instead of '$\phi^{(i)}$ and $\psi^{(i)}$'.
>
> A5. Thank you for pointing out this. We have fixed it in the latest version uploaded to OpenReview.
>
> ----
> Q6.  Sometimes $ \phi $ is in bold and sometimes not (e.g. in Eqs (1) and (3)), not being consistent.
>
> A6. Thank you for pointing out this. We have made all $ \boldsymbol\phi $ in bold to indicate it is a vector. Please refer to the latest version uploaded to OpenReview.
>
> ----
> Q7. The number of current Figure 6 should be Figure 5 instead. The text already refers to it as Figure 5.
>
> A7. We have fixed the referring error in the latex code and the Figure 5 in the revised version is rendered correctly.
>
> ----
> Q8. Explain what ensemble size 0 means in Figure 6b (that is 5b).
>
> A8. Ensemble size being 0 means the no-treatment baseline. We have added the description in the main text in Section 5.4.

---

> > ### Author Response · Authors · 2022-11-18
> > **Part 2 of Response to Reviewer asve**
> >
> > ----
> > Q9. The difference between online and offline distillation.
> >
> > A9. In the online distillation setting, logits of ensembles are generated on the fly when training the target model. This includes the effect of data augmentation. In the offline distillation setting, logits of ensembles are generated from a fixed input before the target training, and thus do not see any training-time data augmentation. The missing data augmentation in the offline distillation setting may contribute to the lower quality results, which is also observed in other tasks involving distillation [C1].
> >
> > [C1] Lucas Beyer, Xiaohua Zhai, Amélie Royer, Larisa Markeeva, Rohan Anil, and Alexander Kolesnikov. Knowledge distillation: A good teacher is patient and consistent. In CVPR, 2022.

---

> ### Author Response · Authors · 2022-11-30
> **Reply**
>
> Thank you again for your time in reviewing our paper. We hope that we have addressed your concerns with our response. Please do not hesitate to let us know if there are any outstanding questions or comments. We would also kindly ask you to consider your evaluation in light of our response.

---

### Author Response · Authors · 2022-11-18
**General Response to All Reviewers**

We thank all reviewers for their constructive feedback and insightful comments. The reviewers agree that the problem that we tackle in this paper is practical (“frequently occurs in production systems” from reviewer V2YQ) and important (reviewer asve). Our theoretical analysis is interesting (reviewer asve and 38Tg) and the simulation studies are clear and supportive (reviewer 38Tg). Our experiment results are extensive (reviewer uaPa). The paper is well written and easy to follow (reviewer asve, V2YQ and uaPa). According to the comments, We have revised the paper where the changes are highlighted in blue. We address individual questions below.

---

### Decision · Program_Chairs · 2023-01-20

**Decision:**

Reject

**Justification For Why Not Higher Score:**

The contributions of the paper are not significant enough to warrant publication.


**Justification For Why Not Lower Score:**

N/A

**Metareview: Summary, Strengths And Weaknesses:**

The paper addresses the task of updating a model in production with a better performing model. One practical aspect is in such situations is to minimize the Negative Flip Rate (NFR): the rate at which the new model makes mistakes on examples that were previously correctly classified.

The paper presents a rather strati-froward combination of prior work: using ensembles to reduct the NFR, and using knowledge distillation to reduct the memory and computational footprint of ensembles. As such, the papers' contribution is of limited significance. A second contribution of the paper is to propose a novel loss for the knowledge distillation procedure that generalized the original LDI loss. However, the authors failed to show that he new generalization is useful. It introduces extra complexity (an extra hyperparameter that needs  to be set) but according to the results provided by the authors during the discussion phase there is no significant performance benefit to be gained.